# Multi-Instance Partial-Label Learning with Margin Adjustment

**Wei Tang**[1,2], **Yin-Fang Yang**[1,2], **Zhaofei Wang**[1,2], **Weijia Zhang**[3], **Min-Ling Zhang**[1,2*]

[1]School of Computer Science and Engineering, Southeast University, Nanjing 210096, China
[2]Key Laboratory of Computer Network and Information Integration (Southeast University),
Ministry of Education, China
[3]School of Information and Physical Sciences, The University of Newcastle,
Callaghan, NSW 2308, Australia
`tangw@seu.edu.cn, yangyf22@gmail.com, wangzf@seu.edu.cn,`
`weijia.zhang@newcastle.edu.au, zhangml@seu.edu.cn`

## Abstract

Multi-instance partial-label learning (MIPL) is an emerging learning framework where each training sample is represented as a multi-instance bag associated with a candidate label set. Existing MIPL algorithms often overlook the margins for attention scores and predicted probabilities, leading to suboptimal generalization performance. A critical issue with these algorithms is that the highest prediction probability of the classifier may appear on a non-candidate label. In this paper, we propose an algorithm named MIPLMA, i.e., *Multi-Instance Partial-Label learning with Margin Adjustment*, which adjusts the margins for attention scores and predicted probabilities. We introduce a margin-aware attention mechanism to dynamically adjust the margins for attention scores and propose a margin distribution loss to constrain the margins between the predicted probabilities on candidate and non-candidate label sets. Experimental results demonstrate the superior performance of MIPLMA over existing MIPL algorithms, as well as other well-established multi-instance learning algorithms and partial-label learning algorithms.

## 1   Introduction

Weakly supervised learning is a powerful strategy for constructing predictive models with limited supervision. Based on the quality and quantity of supervision, Zhou [1] systematically categorizes weak supervision into three types: inexact, inaccurate, and incomplete supervision. Inexact supervision indicates a coarse alignment between instances and labels, which is a common and challenging issue in real-world tasks. *Multi-instance learning (MIL)* [2–8] and *partial-label learning (PLL)* [9–15] are two predominant weekly supervised learning frameworks for learning from samples with inexact supervision in the instance space and the label space, respectively.

Recently, *multi-instance partial-label learning (MIPL)* [16] has been introduced to handle *dual inexact supervision*, where inexact supervision exists in both the instance space and label space. Therefore, MIPL can be seen as a generalized framework of MIL and PLL. In MIPL, a training sample is represented as a multi-instance bag associated with a candidate label set. The candidate label set comprises one true label and the remaining are false positives. The multi-instance bag contains at least one instance corresponding to the true label and does not contain any instance associated with the false positives. Additionally, *positive instances* refer to the instances that belong to the true label, while *negative instances* represent the remaining instances in the bag that are not

---

[*]Corresponding author

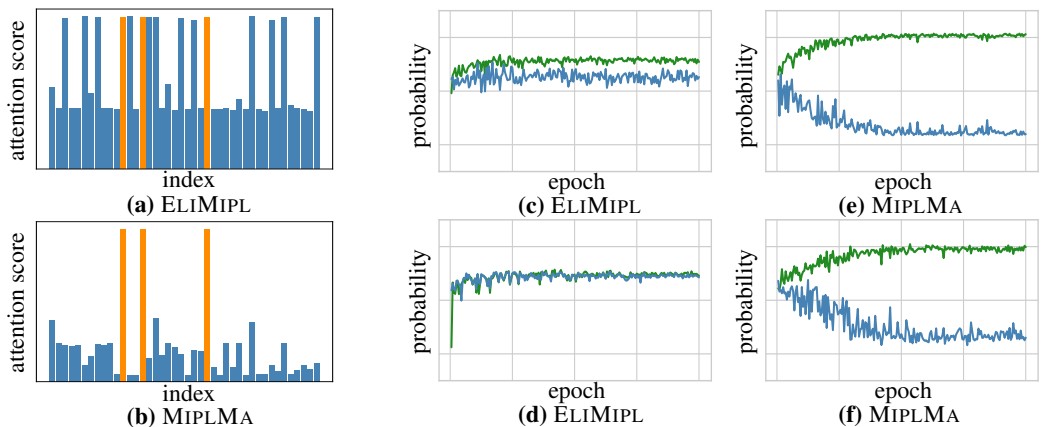

Figure 1: Margin violations in the instance space and the label space. (a) and (b) depict the attention scores of ELIMIPL and MIPLMA for the same test bag in the FMNIST-MIPL dataset. Orange and blue colors indicate attention scores assigned to positive and negative instances, respectively. (c)–(f) show the highest predicted probabilities for candidate labels (green) and non-candidate labels (blue) by ELIMIPL or MIPLMA in the CRC-MIPL-Row dataset. (c) and (e) correspond to the same training bag, while (d) and (f) refer to another training bag.

associated with any label in the label space. During training, the identities of the positive instances and the true label are inaccessible.

Dual inexact supervision widely exists in many tasks. In the classification of histopathological images, an image is frequently partitioned into a multi-instance bag due to its high resolution [17–19, 6] and employing domain experts for providing ground truth labels are costly. As a result, the utilization of crowd-sourced candidate label sets proves to be a valuable strategy in substantially mitigating labeling expenses [20]. To address colorectal cancer classification under dual inexact supervision, Tang et al. [21] have introduced the MIPL algorithm named DEMIPL. This approach employs an attention mechanism to aggregate all instances within a bag into a bag-level feature representation and a disambiguation strategy to identify the true label. Following DEMIPL, ELIMIPL algorithm has been proposed to exploit the label information from both candidate and non-candidate label sets [22]. Additionally, the early MIPL algorithm MIPLGP predicts a bag-level label by aggregating all instance-level labels within the bag without utilizing attention mechanisms [16].

However, existing MIPL algorithms fail to consider the dynamics of the margin between attention scores of positive and negative instances, as well as the margin between the candidate and the non-candidate label sets. These oversights could lead to two major issues. First, the attention scores for positive and negative instances can be quite similar, and in some cases, negative instances may even receive higher attention scores than positive ones, as illustrated in Figure 1(a). Second, the classifier may even assign higher predicted probabilities to non-candidate labels than to candidate labels. Figure 1(c) illustrates a scenario where ELIMIPL assigns predicted probabilities to the candidate labels that are only marginally higher than the non-candidate ones. Furthermore, ELIMIPL may even output lower predicted probabilities for candidate labels than non-candidate ones, as depicted in Figure 1(d). Such erroneous predictions may have serious consequences in applications. For example, in medical image classification, misclassifying images of severe conditions as mild or disease-free may cause patients to miss the opportunity for timely treatment. In this paper, we term this phenomenon as *margin violations*, where the attention scores of negative instances surpass those of positives, or the predicted probabilities for non-candidate labels exceed those for candidate ones. Margin violations occur in both the instance and label spaces, adversely affecting the model's generalization.

To overcome margin violations, we propose a novel end-to-end MIPL algorithm named MIPLMA, i.e., *Multi-Instance Partial-Label learning with Margin Adjustment*. Specifically, to mitigate margin violations in the instance space, we introduce a margin-aware attention mechanism to consolidate each multi-instance bag into a unified feature representation, incorporating dynamic margin adjustments for attention scores. To address margin violations in the label space, we propose a margin distribution loss that adjusts the margin distribution between the model's highest predicted probability for candidate labels and its highest predicted probability for non-candidate labels. In Figure 1(a), MIPLMA allocates

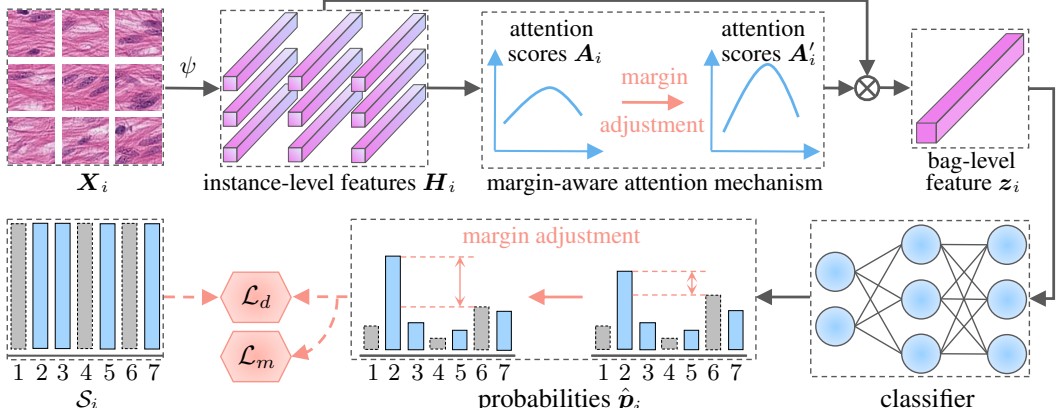

Figure 2: The MIPLMA framework processes an input comprising the multi-instance bag $\boldsymbol{X}_i = \{\boldsymbol{x}_{i,1}, \boldsymbol{x}_{i,2}, \cdots, \boldsymbol{x}_{i,9}\}$ and the candidate label set $\mathcal{S}_i = \{2, 3, 5, 7\}$, where $\mathcal{L}_d$ and $\mathcal{L}_m$ represent the dynamic disambiguation loss and the margin distribution loss, respectively.

higher attention scores to positive instances and enlarges the gap between the attention scores of positive and negative instances. As illustrated in Figure 1(e) and (f), MIPLMA significantly enhances the classifier's highest predicted probability on candidate labels while concurrently reducing the model's highest predicted probability on non-candidate labels. Consequently, our margin adjustment strategy effectively reduces supervision inexactness in both the instance space and the label space.

Our contributions can be summarized as follows: First, we identify the phenomenon of margin violations and adjust the margins in both the instance and label spaces to alleviate this issue. Second, our proposed MIPLMA outperforms state-of-the-art methods significantly. Third, the introduced margin-aware attention mechanism enhances the performance of MIL algorithms, while the margin distribution loss improves the generalization ability of PLL algorithms.

## 2 The Proposed Approach

Formally, we define a MIPL training dataset as $\mathcal{D} = \{(\boldsymbol{X}_i, \mathcal{S}_i) \mid 1 \leq i \leq m\}$, comprising $m$ multi-instance bags and their corresponding candidate label sets. Specifically, each candidate label set $\mathcal{S}_i$ includes one true label, and the remaining are false positives. We denote the instance space as $\mathcal{X} = \mathbb{R}^d$, and the label space as $\mathcal{Y} = \{1, 2, \cdots, k\}$, covering $k$ class labels. The $i$-th bag $\boldsymbol{X}_i = \{\boldsymbol{x}_{i,1}, \boldsymbol{x}_{i,2}, \cdots, \boldsymbol{x}_{i,n_i}\}$ consists of $n_i$ instances in the $d$-dimensional space. Both the candidate label set $\mathcal{S}_i$ and the non-candidate label set $\bar{\mathcal{S}}_i$ are proper subsets of the label space $\mathcal{Y}$ and adhere to the conditions $|\mathcal{S}_i| + |\bar{\mathcal{S}}_i| = |\mathcal{Y}| = k$, where $|\cdot|$ represents the cardinality of a set.

The overall framework of MIPLMA is depicted in Figure 2. Initially, we employ a feature extractor $\psi$ to learn instance-level feature representations $\boldsymbol{H}_i$ within the multi-instance bag $\boldsymbol{X}_i$. Subsequently, we propose a margin-aware attention mechanism with adjustable margins of attention scores to fuse $\boldsymbol{H}_i$ into a unified feature representation $\boldsymbol{z}_i$. Lastly, we utilize a classifier to predict the probabilities $\hat{\boldsymbol{p}}_i$ of the multi-instance bag. To identify the true label from the candidate label set, we introduce the dynamic disambiguation loss $\mathcal{L}_d$ and the margin distribution loss $\mathcal{L}_m$.

### 2.1 Margin Adjustment in the Instance Space

For a given multi-instance bag $\boldsymbol{X}_i = \{\boldsymbol{x}_{i,1}, \boldsymbol{x}_{i,2}, \cdots, \boldsymbol{x}_{i,n_i}\} \in \mathbb{R}^{d \times n_i}$ comprising $n_i$ instances, we utilize a feature extractor $\psi$ to learn instance-level feature representations, which is defined as follows:

$$\boldsymbol{H}_i = \psi(\boldsymbol{X}_i) = \{\boldsymbol{h}_{i,1}, \boldsymbol{h}_{i,2}, \cdots, \boldsymbol{h}_{i,n_i}\}. \tag{1}$$

Here, $\boldsymbol{H}_i \in \mathbb{R}^{l \times n_i}$ represents the instance-level feature representation of the multi-instance bag $\boldsymbol{X}_i$, and $\boldsymbol{h}_{i,j}$ denotes the feature representation of the $j$-th instance in the multi-instance bag $\boldsymbol{X}_i$.

The subsequent step involves computing attention scores for each instance. In MIPL, attention scores of all instances are closely distributed during the early stages of training. However, as

training progresses, attention scores for positive instances gradually become higher than those for negative instances [21]. Due to dual inexact supervision, the attention mechanism struggles to differentiate between positive and negative instances during the initial training phases and calculate their corresponding attention scores. As training continues, the attention mechanism gradually assigns more distinct attention scores to positive and negative instances.

Motivated by this observation, we introduce a margin-aware attention mechanism that dynamically adjusts the margin of attention scores to achieve a closer alignment with the model's training process. The computation of attention scores is given by:

$$\boldsymbol{A}_i = \mathrm{softmax}\left(\boldsymbol{W}^\top \left(\tanh\left(\boldsymbol{W}_1^\top \boldsymbol{H}_i\right) \odot \mathrm{sigm}\left(\boldsymbol{W}_2^\top \boldsymbol{H}_i\right)\right)/\tau^{(t)}\right), \tag{2}$$

where $\boldsymbol{W}^\top$, $\boldsymbol{W}_1^\top$, and $\boldsymbol{W}_2^\top$ are learnable parameters. $\tanh(\cdot)$ and $\mathrm{sigm}(\cdot)$ are the hyperbolic tangent and sigmoid functions, respectively. The operator $\odot$ denotes element-wise multiplication, and $\tau^{(t)}$ denotes the *temperature parameter* of the margin-aware attention mechanism. Specifically, in the early training stages, a larger temperature parameter is employed to smooth the distribution of attention scores, preventing the attention mechanism from assigning high scores to instances that are not unequivocally identified as positive or negative. In the later training stages, a smaller temperature parameter is used to sharpen the distribution of attention scores, thereby widening the gap between attention scores for positive and negative instances. Consequently, throughout the training process, the temperature parameter at the $t$-th epoch is dynamically represented as follows:

$$\tau^{(t)} = \max\{\tau_m, \tau^{(t-1)} * 0.95\}, \tag{3}$$

where $\tau_m$ and $\tau^{(t-1)}$ represent the minimum temperature and the temperature at the $(t-1)$-th epoch, respectively. Therefore, Eq. (3) describes an annealing process for the temperature parameter $\tau^{(t)}$.

For multi-instance bags with varying numbers of positive instances, the distribution of attention scores exhibits variations. Consequently, different multi-instance bags require varying temperature parameters. To address this issue, we introduce the following *normalization operations* for the attention scores:

$$\boldsymbol{A}_i' = \frac{\boldsymbol{A}_i - \bar{\boldsymbol{A}}_i}{\sqrt{\sum_{j=1}^{n_i}(a_{i,j} - \bar{a}_i)^2/(n_i - 1)}}, \tag{4}$$

where $\bar{a}_i = \frac{1}{n_i}\sum_{j=1}^{n_i} a_{i,j}$ is the mean value of the attention score $\boldsymbol{A}_i$ and $\bar{\boldsymbol{A}}_i = [\bar{a}_i, \bar{a}_i, \cdots, \bar{a}_i] \in \mathbb{R}^{1 \times n_i}$. Subsequent to obtaining normalized attention scores, we aggregate the instance-level feature representations to compose the bag-level feature representation $\boldsymbol{z}_i \in \mathbb{R}^l$ in the following manner:

$$\boldsymbol{z}_i = \boldsymbol{H}_i \boldsymbol{A}_i'^\top. \tag{5}$$

We now discuss the theoretical properties of the proposed margin-aware attention mechanism. Based on the definitions of the permutation and permutation invariance (Appendix A), the margin-aware attention mechanism can be seen as the operator $\mathcal{A}$. Then, we have the following theorem:

**Theorem 1.** *The margin-aware attention mechanism is permutation invariant.*

Theorem 1 demonstrates that the margin-aware attention mechanism remains unaffected by the order of instances within multi-instance bags. This property is crucial for algorithms that handle set inputs [23, 4]. The proof is provided in Appendix A.

## 2.2 Margin Adjustment in the Label Space

With the aggregated bag-level feature representation, we utilize a classifier that synergizes dynamic disambiguation loss and margin distribution loss to identify the true label.

The aim of our *dynamic disambiguation loss* is to progressively identify the true labels by calculating the classification loss, as illustrated below:

$$\mathcal{L}_{\mathrm{d}} = -\frac{1}{m}\sum_{i=1}^{m}\sum_{c \in \mathcal{S}_i} p_{i,c}^{(t)} \log(\hat{p}_{i,c}^{(t)}), \tag{6}$$

where $p_{i,c}^{(t)}$ and $\hat{p}_{i,c}^{(t)}$ represent the weight and predicted probability, respectively, on the $c$-th class at the $t$-th iteration. This weight represents the probability that the corresponding candidate label is the true label, which is initialized as follows:

$$p_{i,c}^{(0)} = \left\{ \begin{array}{ll} \frac{1}{|\mathcal{S}_i|} & \text{if } c \in \mathcal{S}_i, \\ 0 & \text{otherwise,} \end{array} \right. \tag{7}$$

where $|\cdot|$ represents the set cardinality. In the $t$-th epoch, we update the weight as:

$$p_{i,c}^{(t)} = \left\{ \begin{array}{ll} \alpha^{(t)} p_{i,c}^{(t-1)} + (1 - \alpha^{(t)}) \dfrac{\hat{p}_{i,c}^{(t)}}{\sum_{c' \in \mathcal{S}_i} \hat{p}_{i,c'}^{(t)}} & \text{if } c \in \mathcal{S}_i, \\ 0 & \text{otherwise,} \end{array} \right. \tag{8}$$

where $\alpha^{(t)} = (T - t)/T$ is a tuning parameter used to balance the update speed of the weight, and $T$ is the maximum number of training epochs.

The dynamic disambiguation loss adjusts the classifier's predicted probabilities for the candidate labels, without affecting the probabilities assigned to non-candidate labels. As illustrated in Figure 1(d), this circumstance may result in the classifier assigning its highest predicted probability to a non-candidate label instead of a candidate label, i.e., margin violations. To mitigate potential issues in model generalization, it is crucial to maintain a significant margin between the highest predicted probabilities for the candidate and non-candidate labels. Therefore, we propose the *margin loss* to maximize the margin between the highest predicted probability on the candidate label set and on the non-candidate label set, as shown below:

$$\mathcal{L}_{\text{ml}} = \frac{1}{m} \sum_{i=1}^{m} \{ 1 - (\max_{c \in \mathcal{S}_i} \hat{p}_{i,c}^{(t)} - \max_{\bar{c} \in \bar{\mathcal{S}}_i} \hat{p}_{i,\bar{c}}^{(t)}) \}, \tag{9}$$

where $\max_{c \in \mathcal{S}_i} \hat{p}_{i,c}^{(t)}$ and $\max_{\bar{c} \in \bar{\mathcal{S}}_i} \hat{p}_{i,\bar{c}}^{(t)}$ are the highest predicted probabilities on the candidate label set and the non-candidate label set, respectively. However, only considering the mean margin cannot effectively address margin violations, thus affecting the performance. Some recent studies have shown that the model performance can be enhanced by maximizing the margin mean and minimizing the margin variance simultaneously [24–26]. Therefore, we employ two statistics of the margins i.e., the margin mean and the margin variance, to adjust the margin distribution. Specifically, we can maximize the margin mean and minimize the margin variance between the highest predicted probability on the candidate label set and on the non-candidate label set simultaneously by minimizing the following *margin distribution loss*:

$$\mathcal{L}_{\text{m}} = \frac{\mathcal{M}\{\phi_1, \phi_2, \cdots, \phi_m\}}{1 - \sqrt{\mathcal{V}\{\phi_1, \phi_2, \cdots, \phi_m\}}}, \tag{10}$$

where $\phi_i = \{1 - (\max_{c \in \mathcal{S}_i} \hat{p}_{i,c}^{(t)} - \max_{\bar{c} \in \bar{\mathcal{S}}_i} \hat{p}_{i,\bar{c}}^{(t)})\}$ refers to the margin loss of the $i$-th multi-instance bag. $\mathcal{M}\{\cdot\}$ and $\mathcal{V}\{\cdot\}$ are the mean and the variance of the margin loss, respectively.

During training, the full loss is represented as the weighted sum of the dynamic disambiguation loss and the margin distribution loss, as expressed below:

$$\mathcal{L} = \mathcal{L}_{\text{d}} + \lambda \mathcal{L}_{\text{m}}, \tag{11}$$

where $\lambda$ represents a hyperparameter.

## 3 Experiments

### 3.1 Experimental Configurations

#### 3.1.1 Datasets

Following the experimental setup of DEMIPL [21], we utilize four MIPL benchmark datasets and one real-world dataset. The four benchmark datasets encompass MNIST-MIPL, FMNIST-MIPL, Birdsong-MIPL, and SIVAL-MIPL, spanning diverse application domains such as image analysis and biology [27–30]. Additionally, the real-world CRC-MIPL dataset is annotated by crowdsourced workers for

Table 1: Characteristics of the benchmark and real-world MIPL datasets.

| Dataset | #bag | #ins | max. #ins | min. #ins | avg. #ins | #dim | #class | avg. #CLs |
|---------|------|------|-----------|-----------|-----------|------|--------|-----------|
| MNIST-MIPL | 500 | 20664 | 48 | 35 | 41.33 | 784 | 5 | 2, 3, 4 |
| FMNIST-MIPL | 500 | 20810 | 48 | 36 | 41.62 | 784 | 5 | 2, 3, 4 |
| Birdsong-MIPL | 1300 | 48425 | 76 | 25 | 37.25 | 38 | 13 | 2, 3, 4 |
| SIVAL-MIPL | 1500 | 47414 | 32 | 31 | 31.61 | 30 | 25 | 2, 3, 4 |
| C-Row | 7000 | 56000 | 8 | 8 | 8 | 9 | 7 | 2.08 |
| C-SBN | 7000 | 63000 | 9 | 9 | 9 | 15 | 7 | 2.08 |
| C-KMeans | 7000 | 30178 | 6 | 3 | 4.311 | 6 | 7 | 2.08 |
| C-SIFT | 7000 | 175000 | 25 | 25 | 25 | 128 | 7 | 2.08 |
| C-R34-16 | 7000 | 112000 | 16 | 16 | 16 | 1000 | 7 | 2.08 |
| C-R34-25 | 7000 | 175000 | 25 | 25 | 25 | 1000 | 7 | 2.08 |

colorectal cancer classification. The previous works [21, 22] employ four distinct types of multi-instance features and consists of four sub-datasets: CRC-MIPL-Row (C-Row), CRC-MIPL-SBN (C-SBN), CRC-MIPL-KMeansSeg (C-KMeans), and CRC-MIPL-SIFT (C-SIFT). These multi-instance features are generated via four image bag generators [31], i.e., Row, single blob with neighbors (SBN), k-means segmentation (KMeansSeg), and scale-invariant feature transform (SIFT), respectively. Besides these multi-instance features, we are the first to employ the ResNet [32] to learn the multi-instance features of CRC-MIPL dataset. Specifically, we partition each image into $N$ non-overlapping patches, treating each patch as an instance. Subsequently, the ResNet-34 is employed to acquire feature representations for each patch, resulting in feature representations of dimension 1000 for each patch. In our experiments, the $N$ is 16 and 25, and the resulting datasets are CRC-MIPL-ResNet-34-16 (C-R34-16) and CRC-MIPL-ResNet-34-25 (C-R34-25).

The characteristics of the dataset are detailed in Table 1. It provides the number of multi-instance bags and total instances, denoted as *#bag* and *#ins*, respectively. Furthermore, we employ *max. #ins*, *min. #ins*, and *avg. #ins* to express the maximum, minimum, and average instance count within all bags. The dimensionality of each instance-level feature representation is indicated by *#dim*. *#class* and *avg. #CLs* denote the length of the label space and the average length of candidate label sets, respectively. For a comprehensive performance assessment, we vary the number of false positive labels on the benchmark datasets, represented as $r$ ($|\mathcal{S}_i| = r + 1$).

### 3.1.2 Comparative Algorithms

We conduct a comprehensive comparison of MIPLMA with a wide variety of baselines, covering MIPL, PLL, and MIL algorithms. For MIPL algorithms, we compare with MIPLGP [16], DEMIPL [21], and ELIMIPL [22]. In our evaluation, we incorporate seven PLL algorithms, featuring five deep-learning-based approaches: PRODEN [33], RC [34], LWS [35], CAVL [11], and POP [36], one feature-aware disambiguation algorithm, PL-AGGD [37], and two margin-based algorithms, M3PL [38] and PL-SVM [39]. Furthermore, our comparison encompasses seven MIL algorithms. Three of the MIL algorithms are Gaussian processes-based: VWSGP [40], VGPMIL [41], and LM-VGPMIL [41]. Additionally, a variational autoencoder-based algorithm, MIVAE [42], and three attention-based algorithms: ATTEN [4], ATTEN-GATE [4], and LOSS-ATTEN [43], are included.

The deep-learning-based PLL algorithms [33–35, 11] can be equipped with either the linear model or multi-layer perceptrons (MLP) as backbone networks. Results obtained from the linear model are presented in the main body of the paper, while additional experiment results are detailed in the Appendix C. Parameters for all compared baselines have been meticulously tuned, drawing from recommendations in the original literature or refined through our pursuit of improved performance.

### 3.1.3 Implementation

We implement MIPLMA using PyTorch [44] and conduct training with a single NVIDIA Tesla V100 GPU. Employing the stochastic gradient descent (SGD) optimizer, we set the momentum value to 0.9 with a weight decay of 0.0001. To learn the instance-level features, we employ a two-layer convolutional neural network and a fully connected network for the MNIST-MIPL and FMNIST-MIPL datasets. Since the features of the Birdsong-MIPL, and SIVAL-MIPL datasets are preprocessed, we only employ a fully connected network to learn the feature representations. For the CRC-MIPL dataset, the feature extractor is one of the four image bag generators or ResNet-34, followed by a fully

Table 2: The classification accuracies (mean±std) of MIPLMA and comparative algorithms on the benchmark datasets with the varying numbers of false positive labels ($r \in \{1, 2, 3\}$).

| Algorithm | $r$ | MNIST-MIPL | | FMNIST-MIPL | | Birdsong-MIPL | | SIVAL-MIPL | |
|---|---|---|---|---|---|---|---|---|---|
| MIPLMA | 1 | .985±.010 | | **.915±.016** | | **.776±.020** | | **.703±.026** | |
| | 2 | .979±.014 | | **.867±.028** | | **.762±.015** | | **.668±.031** | |
| | 3 | **.749±.103** | | .654±.055 | | **.746±.013** | | **.627±.024** | |
| ELIMIPL | 1 | **.992±.007** | | .903±.018 | | .771±.018 | | .675±.022 | |
| | 2 | **.987±.010** | | .845±.026 | | .745±.015 | | .616±.025 | |
| | 3 | .748±.144 | | **.702±.055** | | .717±.017 | | .600±.029 | |
| DEMIPL | 1 | .976±.008 | | .881±.021 | | .744±.016 | | .635±.041 | |
| | 2 | .943±.027 | | .823±.028 | | .701±.024 | | .554±.051 | |
| | 3 | .709±.088 | | .657±.025 | | .696±.024 | | .503±.018 | |
| MIPLGP | 1 | .949±.016 | | .847±.030 | | .716±.026 | | .669±.019 | |
| | 2 | .817±.030 | | .791±.027 | | .672±.015 | | .613±.026 | |
| | 3 | .621±.064 | | .670±.052 | | .625±.015 | | .569±.032 | |
| | | Mean | MaxMin | Mean | MaxMin | Mean | MaxMin | Mean | MaxMin |
| PRODEN | 1 | .605±.023 | .508±.024 | .697±.042 | .424±.045 | .296±.014 | .387±.014 | .219±.014 | .316±.019 |
| | 2 | .481±.036 | .400±.037 | .573±.026 | .377±.040 | .272±.019 | .357±.012 | .184±.014 | .287±.024 |
| | 3 | .283±.028 | .345±.048 | .345±.027 | .309±.058 | .211±.013 | .336±.012 | .166±.017 | .250±.018 |
| RC | 1 | .658±.031 | .519±.028 | .753±.042 | .731±.027 | .362±.015 | .390±.014 | .279±.011 | .306±.023 |
| | 2 | .598±.033 | .469±.035 | .649±.028 | .666±.027 | .335±.011 | .371±.013 | .258±.017 | .288±.021 |
| | 3 | .392±.033 | .380±.048 | .401±.063 | .524±.034 | .298±.009 | .363±.010 | .237±.020 | .267±.020 |
| LWS | 1 | .463±.048 | .242±.042 | .726±.031 | .435±.049 | .265±.010 | .225±.038 | .240±.014 | .289±.017 |
| | 2 | .209±.028 | .239±.048 | .720±.025 | .406±.040 | .254±.010 | .207±.034 | .223±.008 | .271±.014 |
| | 3 | .205±.013 | .218±.017 | .579±.041 | .318±.064 | .237±.005 | .216±.029 | .194±.026 | .244±.023 |
| CAVL | 1 | .596±.074 | .481±.030 | .728±.047 | .370±.025 | .370±.012 | .354±.015 | .260±.013 | .251±.023 |
| | 2 | .412±.039 | .389±.027 | .586±.055 | .264±.037 | .316±.017 | .335±.008 | .237±.001 | .216±.011 |
| | 3 | .315±.020 | .292±.032 | .353±.025 | .265±.025 | .272±.031 | .313±.017 | .197±.014 | .175±.020 |
| POP | 1 | .657±.033 | .511±.032 | .799±.032 | .409±.044 | .383±.009 | .388±.015 | .295±.012 | .316±.015 |
| | 2 | .585±.045 | .438±.037 | .725±.025 | .395±.028 | .348±.011 | .360±.018 | .278±.020 | .296±.020 |
| | 3 | .335±.022 | .362±.034 | .619±.049 | .324±.032 | .312±.012 | .345±.014 | .251±.021 | .255±.010 |
| PL-AGGD | 1 | .671±.027 | .527±.035 | .743±.026 | .391±.040 | .353±.019 | .383±.014 | .355±.015 | .397±.028 |
| | 2 | .595±.036 | .439±.020 | .677±.028 | .371±.037 | .314±.018 | .372±.020 | .315±.019 | .360±.029 |
| | 3 | .380±.032 | .321±.043 | .474±.057 | .327±.028 | .296±.015 | .344±.011 | .286±.018 | .328±.023 |

connected network. The initial learning rate is chosen from the set $\{0.01, 0.05\}$ and coupled with a cosine annealing technique. We set the number of epochs to 100 for benchmark datasets and 200 for the CRC-MIPL dataset. The weight of the margin distribution loss is chosen from the set $\{0.01, 0.05, 0.1, 0.5, 1, 3, 5\}$ for all datasets. For the annealing process of the temperature parameter, the initial temperature parameter $\tau^{(0)} = 5$. Additionally, $\tau_m = 0.1$ and $\tau_m = 0.5$ are used for benchmark datasets and the CRC-MIPL dataset, respectively. The dataset partitioning method aligns with that of DEMIPL [21] and ELIMIPL [22]. We execute ten random train/test splits, maintaining a ratio of $7:3$. Mean accuracies and standard deviations from these ten runs are reported. The code of MIPLMA can be found at `https://github.com/tangw-seu/MIPLMA`.

### 3.2 Comparison with MIPL and PLL Algorithms

Since PLL algorithms can not directly handle the multi-instance bags, we utilize two data degradation strategies: the *Mean* strategy and the *MaxMin* strategy [16]. The former computes the average feature values across all instances within a bag for producing a bag-level feature representation. The latter identifies both the maximum and minimum feature values for each dimension among instances within a multi-instance bag and concatenates these values to form a bag-level feature representation.

#### 3.2.1 Results on the Benchmark Datasets

Table 2 provides a comprehensive comparison of the results achieved by MIPLMA, three MIPL algorithms (ELIMIPL [22], DEMIPL [21], and MIPLGP [16]), five deep-learning-based PLL algorithms (PRODEN [33], RC [34], LWS [35], CAVL [11], and POP [36]) with linear model, and the feature-aware disambiguation PLL algorithm (PL-AGGD [37]). The evaluation is conducted on benchmark datasets with varying numbers of false positive labels ($r \in \{1, 2, 3\}$).

Notably, MIPLMA consistently exhibits higher average accuracy than the three MIPL algorithms in 33 out of 36 cases. For the methods based on the embedding space paradigm, MIPLMA demonstrates superior performance compared to ELIMIPL and DEMIPL in 21 out of 24 cases. Compared to MIPLGP that follows the instance space paradigm, MIPLMA achieves higher average accuracies than it in all cases. Specifically, on the SIVAL-MIPL dataset, the average accuracies of MIPLGP consistently

Table 3: The classification accuracies (mean±std) of MIPLMA and comparative algorithms on the real-world datasets. – means unavailable due to computational limitations.

| Algorithm | C-Row | | C-SBN | | C-KMeans | | C-SIFT | |
|---|---|---|---|---|---|---|---|---|
| MIPLMA | **.444±.010** | | **.526±.009** | | **.557±.010** | | **.553±.009** | |
| ELIMIPL | .433±.008 | | .509±.007 | | .546±.012 | | .540±.010 | |
| DEMIPL | .408±.010 | | .486±.014 | | .521±.012 | | .532±.013 | |
| MIPLGP | .432±.005 | | .335±.006 | | .329±.012 | | – | |
| | Mean | MaxMin | Mean | MaxMin | Mean | MaxMin | Mean | MaxMin |
| PRODEN | .365±.009 | .401±.007 | .392±.008 | .447±.011 | .233±.018 | .265±.027 | .334±.029 | .291±.011 |
| RC | .214±.011 | .227±.012 | .242±.012 | .338±.010 | .226±.009 | .208±.007 | .209±.007 | .246±.008 |
| LWS | .291±.010 | .299±.008 | .310±.006 | .382±.009 | .237±.008 | .247±.005 | .270±.007 | .230±.007 |
| CAVL | .312±.043 | .368±.054 | .364±.066 | .503±.025 | .286±.062 | .311±.038 | .329±.033 | .274±.018 |
| POP | .383±.010 | .393±.015 | .439±.009 | .438±.010 | .385±.016 | .279±.016 | .326±.013 | .278±.040 |
| PL-AGGD | .412±.008 | .460±.008 | .480±.005 | .524±.008 | .358±.008 | .434±.009 | .363±.012 | .285±.009 |

exceed those of DEMIPL. However, the average accuracies of MIPLMA surpass all algorithms on the SIVAL-MIPL dataset, thus highlighting the effectiveness of MIPLMA.

Additionally, MIPLMA significantly outperforms PLL algorithms in all cases. For relatively simple datasets such as MNIST-MIPL and FMNIST-MIPL, the PLL algorithms demonstrate satisfactory performance. However, with the increasing complexity of datasets, as observed in Birdsong-MIPL and SIVAL-MIPL, the effectiveness of the PLL algorithms noticeably diminished. On MNIST-MIPL and FMNIST-MIPL, the Mean strategy generally outperforms the MaxMin strategy. Conversely, on Birdsong-MIPL and SIVAL-MIPL, the MaxMin strategy tends to yield superior results in most cases compared to the Mean strategy. Hence, the two data degradation strategies do not uniformly outperform each other but have their respective advantages. The selection of the degradation strategy is dependent on the characteristics of the dataset. For simpler datasets, a straightforward Mean strategy may suffice, while for more complex datasets, a sophisticated MaxMin strategy may be preferable.

### 3.2.2 Results on the Real-World Datasets

Table 3 presents a detailed comparison of results on the CRC-MIPL dataset. Our method, MIPLMA, demonstrates superior performance in all 11 cases when compared to ELIMIPL [22], DEMIPL [21], and MIPLGP [16]. In terms of the PLL algorithm, MIPLMA also achieves superior accuracies in all cases. While the PLL algorithms yield satisfactory results on relatively simple datasets like CRC-MIPL-Row and CRC-MIPL-SBN, their performances noticeably deteriorate when handling more complex datasets such as CRC-MIPL-KMeans and CRC-MIPL-SIFT.

Moreover, both MIPLMA and ELIMIPL demonstrate significantly better performance on the CRC-MIPL-KMeans and CRC-MIPL-SIFT datasets compared to the CRC-MIPL-Row and CRC-MIPL-SBN datasets. However, this trend is reversed for MIPLGP and the PLL algorithms. We attribute this discrepancy to the incapacity of these algorithms to effectively model complex features. Particularly, the limitations of the PLL algorithms become more apparent when dealing with complex MIPL data. Therefore, there is an urgent need to devise more effective MIPL algorithms.

### 3.2.3 Results of the CRC-MIPL Dataset with Deep Features

Tang et al. [21] have introduced the CRC-MIPL dataset, extracting multi-instance features using four hand crafted image bag generators [31]. Both DEMIPL [21] and ELIMIPL [22] were evaluated using these multi-instance features in the literature. In this study, we investigate CRC-MIPL with neural network generated features and employ ResNet to learn deep multi-instance features from the CRC-MIPL dataset. The resulting datasets are referred to as C-R34-16 and C-R34-25.

Table 4 illustrates the classification accuracies of MIPLMA, ELIMIPL, and DEMIPL on the CRC-MIPL dataset with deep multi-instance features. From the experimental results, two key observations emerge: (a) ResNet-34-based features outperform those generated by image bag generators in terms of classification performance. (b) When learning multi-instance features with ResNet-34, dividing an image into 25 instances results in a more discriminative feature representation compared to using 16 instances.

Table 4: The classification accuracies (mean±std) on the CRC-MIPL dataset with deep multi-instance features.

| Algorithm | C-R34-16 | C-R34-25 |
|---|---|---|
| MIPLMA | **.631±.008** | **.685±.011** |
| ELIMIPL | .628±.009 | .663±.009 |
| DEMIPL | .625±.008 | .650±.010 |

Table 5: The classification accuracies (mean±std) of MAAM, ATTEN, and ATTEN-GATE on the MIL datasets with bag-level true labels.

| Algorithm | MNIST-MIPL (MIL) | FMNIST-MIPL (MIL) |
|---|---|---|
| MAAM | **.991±.006** | **.930±.021** |
| ATTEN | .962±.010 | .859±.032 |
| ATTEN-GATE | .971±.017 | .847±.037 |

Table 6: The classification accuracies (mean±std) of PRODEN-MA and PRODEN on the Kuzushiji-MNIST dataset with varying flipping probability $q$.

| Algorithm | $q = 0.1$ | $q = 0.3$ | $q = 0.5$ | $q = 0.7$ | $q = 0.9$ |
|---|---|---|---|---|---|
| PRODEN-MA | **.932±.001** | **.926±.002** | **.914±.002** | **.892±.001** | **.816±.008** |
| PRODEN | .906±.002 | .900±.001 | .884±.005 | .876±.010 | .772±.017 |

In summary, feature representations learned by the deep feature extractor ResNet-34 exhibit higher discriminative capacity compared to those generated by image bag generators. Our model consistently achieves the highest classification accuracy among these three MIPL algorithms, especially on the C-R34-25 dataset. These observations suggest that our model not only achieves the highest classification accuracy on traditional features but also handles deep features better.

### 3.3 Effectiveness of the Margin Adjustment

To assess the effectiveness of margin adjustment, we introduce three variants of MIPLMA. MIPL-MAINS denotes the margin adjustment of attention scores exclusively, with $\lambda$ in Eq. (11) set to 0. MIPL-MALAB signifies the margin adjustment of predicted probabilities only, where $\tau^{(t)}$ in Eq. (2) is assigned 1 for $t = 1, 2, \cdots, T$. MIPL-WOMA indicates no margin adjustments of attention scores or predicted probabilities, with $\lambda = 0$ and $\tau^{(t)} = 1$ for $t = 1, 2, \cdots, T$.

Figure 3 demonstrates that MIPLMA consistently outperforms its three variants, proving that margin adjustment in the instance and label spaces can significantly enhance model performance. Additionally, adjusting the margin in the label space yields better results than in the instance space, validating the effectiveness of our proposed margin distribution loss. From another perspective, adjusting the margin of predicted probabilities directly impacts classi-

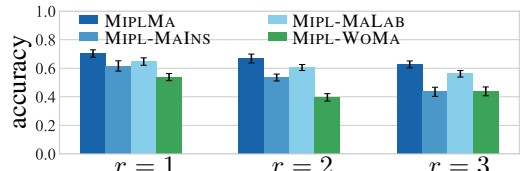

Figure 3: The classification accuracies (mean and std) of MIPLMA with the three variants on the SIVAL-MIPL dataset ($r \in \{1, 2, 3\}$).

fication accuracies, whereas adjusting the margin of attention scores affects the bag-level feature representations, thereby influencing classification accuracies. Consequently, MIPL-MALAB demonstrates superior performance than MIPL-MAINS. By simultaneously adjusting the margins in both the instance and label spaces, i.e., MIPLMA, optimal results can be achieved. This underscores the effectiveness of our margin adjustment strategy in dealing with the inexact supervision of MIPL.

### 3.4 Margin Adjustment for MIL and PLL Algorithms

In MIPLMA, the margin adjustments for attention scores and predicted probabilities reduce the supervision inexactness in the instance space and the label space, respectively. MIPL is a generalized framework of MIL and PLL. Therefore, this raises a pertinent question: can margin adjustment enhance the performance of MIL and PLL algorithms?

To answer this question, we propose a MIL algorithm named MAAM that is a simplified variant of MIPLMA. We compare MAAM with two classical MIL methods incorporating attention mechanisms, namely ATTEN [4] and ATTEN-GATE [4], on the MNIST-MIPL (MIL) and FMNIST-MIPL (MIL) datasets. During training, we use only the features of multi-instance bags and their corresponding bag-level true labels. The parameters of MAAM for the two datasets are as follows: learning rates of 0.005 for MNIST-MIPL and 0.01 for FMNIST-MIPL. Additionally, $\tau^{(0)} = 5$ and $\tau_m = 0.1$ for both datasets. Table 5 presents the classification accuracies over ten runs, indicating that MAAM outperforms both ATTEN and ATTEN-GATE, particularly on the FMNIST-MIPL dataset. These results demonstrate the effectiveness of MAAM, confirming its superior performance.

Moreover, we equip the PLL algorithm PRODEN with the margin distribution loss, resulting in the variant PRODEN-MA. Table 6 presents the classification accuracies of PRODEN-MA and PRODEN on the Kuzushiji-MNIST dataset with varying flipping probability $q \in \{0.1, 0.3, 0.5, 0.7, 0.9\}$. The only difference between PRODEN-MA and PRODEN lies in that PRODEN-MA includes the margin distribution loss with the weight of $1$. We employ MLP as the backbone network for both PRODEN-MA and PRODEN, and keep all other parameters consistent. Experimental results indicate that PRODEN-MA outperforms PRODEN across all scenarios. Notably, under higher disambiguation difficulty, i.e., $q = 0.9$, the superiority of PRODEN-MA is more pronounced.

In summary, adjusting the margins of attention scores improves classification performance for MIL algorithms. Similarly, margin adjustment in the label space enhances the performance of PLL algorithms, particularly in challenging disambiguation scenarios.

## 4 Conclusion

This paper investigates the margin adjustments in both the instance space and the label space for MIPL. We propose MIPLMA, which incorporates a margin-aware attention mechanism and a margin distribution loss to adjust the margins for attention scores and predicted probabilities, respectively. Experimental results on the benchmark and real-world datasets illustrate the superiority of our proposed MIPLMA algorithm over a diverse set of baselines, encompassing MIPL, PLL, and MIL algorithms. Specifically, MIPLMA achieves superior performances compared to baselines in $96.4\%$ of cases. These results underscore the effectiveness and significance of our margin adjustment strategy.

However, MIPLMA has several limitations. First, similar to other attention-based MIL and MIPL methods, it cannot process multiple multi-instance bags simultaneously. Second, MIPLMA demonstrates a slight overfitting problem on the relatively simple MNIST-MIPL dataset. Third, MIPLMA is not suitable for instance-level classification tasks. In the future, we will delve into designing MIPL algorithms capable of instance-level classification and parallel algorithms that can handle multiple multi-instance bags concurrently.

## Acknowledgements

The authors wish to thank the anonymous reviewers for their helpful comments and suggestions. This work was supported by the National Science Foundation of China (62225602) and the Big Data Computing Center of Southeast University.

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

# Multi-Instance Partial-Label Learning with Margin Adjustment
# (Appendix)

## A  Proof of Theorem 1

For notation simplicity, we denote the instance-level features $H_i$ as $H$ for the rest of this section. First, we definite the permutation and permutation invariant as follows.

**Definition 1.** *(Permutation).* *We characterize $\mathcal{Q} : \mathbb{R}^{l \times n} \to \mathbb{R}^{l \times n}$ as a permutation operation, denoted by $\mathcal{Q}(H) = HQ$, where $Q \in \mathbb{R}^{n \times n}$ is a permutation matrix employed to rearrange the instance order within $H$. Specifically, $Q$ is an orthogonal matrix, ensuring $Q^\top Q = I$.*

**Definition 2.** *(Permutation Invariant).* *An operator $\mathcal{A} : \mathbb{R}^{l \times n} \to \mathbb{R}^l$ is permutation invariant concerning the instance order in $H$ if $\mathcal{Q}(\mathcal{A}(H)) = \mathcal{Q}(H)$ is satisfied [23].*

**Theorem 1.** *The margin-aware attention mechanism is permutation invariant.*

*Proof.* Let $H = \{h_1, h_2, \ldots, h_n\}$ denote the instance-level feature representations of the multi-instance bag $X$. We rewrite the computation of attention scores in Eq. (2) as follows:

$$
\begin{aligned}
A &= \mathcal{AS}(H) \\
&= \text{softmax}(W^\top(\tanh(W_1^\top H) \odot \text{sigm}(W_2^\top H))/\tau) \\
&= \text{softmax}(\xi(H)/\tau).
\end{aligned}
\tag{A1}
$$

The normalization operations of attention scores and the aggregated bag-level feature representation are shown below:

$$
A' = \Psi(A) = \Psi(\mathcal{AS}(H)),
\tag{A2}
$$

$$
\mathcal{A}(H) = HA' = H\Psi(\mathcal{AS}(H))^\top,
\tag{A3}
$$

where $\Psi(\cdot)$ represents the function of the normalization operations and $A'$ signifies the normalized attention scores.

Given the permutation matrix $Q$ satisfying $Q^\top Q = I$, we obtain the permuted feature $\mathcal{Q}(H) = HQ$. Firstly, the attention scores are computed as follows:

$$
\begin{aligned}
\mathcal{AS}(HQ) &= \text{softmax}(\xi(HQ)/\tau) \\
&= \text{softmax}(W^\top(\tanh(W_1^\top HQ) \odot \text{sigm}(W_2^\top HQ))/\tau) \\
&= \text{softmax}(W^\top(\tanh(W_1^\top H)Q \odot \text{sigm}(W_2^\top H)Q)/\tau) \\
&= \text{softmax}(W^\top(\tanh(W_1^\top H) \odot \text{sigm}(W_2^\top H))Q/\tau) \\
&= \text{softmax}(\xi(H)Q/\tau) \\
&= \text{softmax}(\xi(H)/\tau)Q \\
&= \mathcal{AS}(H)Q.
\end{aligned}
\tag{A4}
$$

As depicted in Equation (4), in the normalization operations, neither the denominator nor the mean value of attention scores is affected by the permutation matrix $Q$. Therefore, we can derive the following equation:

$$
\Psi(\mathcal{AS}(HQ)) = \Psi(\mathcal{AS}(H)Q) = \Psi(\mathcal{AS}(H))Q.
\tag{A5}
$$

Subsequently, the resultant representation of the aggregated bag-level features is expressed as:

$$
\begin{aligned}
\mathcal{A}(HQ) &= HQ\Psi(\mathcal{AS}(HQ))^\top \\
&= HQ\Psi(\mathcal{AS}(H)Q)^\top \\
&= HQQ^\top\Psi(\mathcal{AS}(H))^\top \\
&= H\Psi(\mathcal{AS}(H))^\top \\
&= \mathcal{A}(H).
\end{aligned}
\tag{A6}
$$

Therefore, the margin-aware attention mechanism is permutation invariant. $\qquad \square$

---

**Algorithm 1** $Y_* = \text{MIPLMA}\ (\mathcal{D}, \lambda, T, \boldsymbol{X}_*)$

---

**Inputs**:

$\mathcal{D}$: the MIPL training set $\{(\boldsymbol{X}_i, \mathcal{S}_i) \mid 1 \le i \le m\}$, where $\boldsymbol{X}_i = \{\boldsymbol{x}_{i,1}, \boldsymbol{x}_{i,2}, \cdots, \boldsymbol{x}_{i,n_i}\}$, $\boldsymbol{x}_{i,j} \in \mathcal{X}$, $\mathcal{X} = \mathbb{R}^d$, $\mathcal{S}_i \subset \mathcal{Y}$, $\mathcal{Y} = \{1, 2, \cdots, k\}$

$\lambda$: the weight for the maximum-margin disambiguation loss

$T$: the maximum number of training epochs

$\boldsymbol{X}_*$: the unseen multi-instance bag with $n_*$ instances

**Outputs**:

$\boldsymbol{Y}_*$: the predicted label for $\boldsymbol{X}_*$

**Process**:

 1: Initialize uniform the weights on candidate label set $p_{i,c}^{(0)}$ as stated by Eq. (7)
 2: **for** $t = 1$ to $T$ **do**
 3:    Shuffle training set $\mathcal{D}$ into $B$ mini-batches
 4:    **for** $b = 1$ to $B$ **do**
 5:       Extract the instance-level features according to Eq. (1)
 6:       Calculate the attention scores with the temperature $\tau^{(t)}$ as stated by Eqs. (2) and (3)
 7:       Normalize the attention scores as stated by Eq. (4)
 8:       Aggregate the instance-level feature representations into the bag-level feature representation according to Eq. (5)
 9:       Classify the multi-instance bag with predicted probabilities.
10:       Update the weights $p_{i,c}^{(t)}$ according to Eq. (8)
11:    **end for**
12:    Calculate the full loss $\mathcal{L}$ according to Eq. (11)
13:    Update the model $\Phi$ by the optimizer
14: **end for**
15: Extract the instance-level features of $\boldsymbol{X}_*$ according to Eq. (1)
16: Calculate the attention scores and map the instance-level features into a single vector representation $\boldsymbol{z}_*$ according to Eqs. (2), (3), (4), and (5)
17: Return $Y_* = \underset{c \in \mathcal{Y}}{\arg\max}\ \hat{p}_{*,c}$

---

## B    Pseudo-Code of MIPLMA

Algorithm 1 describes the complete procedure of MIPLMA. First, the algorithm uniformly initializes the weights on the candidate label set (Step 1). In each epoch, the training set is divided into multiple mini-batches (Step 3). Then, instance-level feature representations are extracted for each mini-batch and aggregated into bag-level feature representation (Steps 5-8). The subsequent step involves classify the multi-instance bag and updating the weights on the candidate label set (Steps 9-10). Last, the full loss is calculated, and the model is updated (Steps 12-13). For an unseen multi-instance bag, instance-level feature representations are learned and aggregated into a bag-level feature representation via the feature extractor and the margin-aware attention mechanism, respectively (Steps 15-16). The predicted label is the category corresponding to the highest prediction probability (Step 17).

## C    Additional Experiment Results

### C.1    Comparison with Margin-based PLL Algorithms

In the field of PLL, there exist several disambiguation algorithms that rely on the maximum margin criteria, with PL-SVM and M3PL emerging as notable algorithms. PL-SVM achieves disambiguation by maximizing the margin mean between the highest prediction probability on the candidate label set and that on the non-candidate label set. In contrast, M3PL maximizes the margin mean between the highest and second-highest prediction probabilities of the classifier for disambiguation. Consequently, we conduct a comparative analysis of classification performance using the MIPLMA algorithm against PL-SVM and M3PL. Figures A1 and present the average accuracy and standard deviation of the three algorithms over ten runs on benchmark and real-world datasets, with Mean and MaxMin representing the corresponding data degradation strategies.

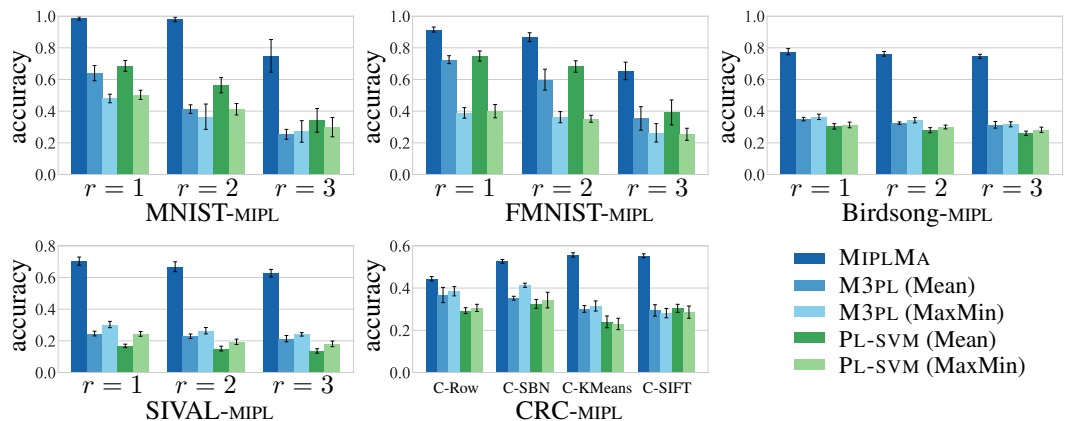

Figure A1: The classification accuracies (mean and std) of MIPLMA, M3PL, and PL-SVM.

Table A1: The classification accuracies (mean±std) of MIPLMA and comparative MIL algorithms on the benchmark datasets with one false positive candidate label ($r = 1$).

| Algorithm | MNIST-MIPL | FMNIST-MIPL | Birdsong-MIPL | SIVAL-MIPL |
|---|---|---|---|---|
| MIPLMA | **.985±.010** | **.915±.016** | **.776±.020** | **.703±.026** |
| VWSGP | .413±.043 | .416±.045 | .271±.045 | .050±.013 |
| VGPMIL | .448±.052 | .481±.030 | .072±.038 | .042±.006 |
| LM-VGPMIL | .483±.050 | .465±.024 | .079±.055 | .046±.008 |
| MIVAE | .692±.221 | .603±.212 | .142±.191 | .098±.133 |
| ATTEN | .499±.010 | .408±.069 | .134±.020 | .100±.021 |
| ATTEN-GATE | .504±.078 | .357±.101 | .161±.026 | .110±.012 |
| LOSS-ATTEN | .842±.051 | .777±.048 | .537±.025 | .320±.028 |

MIPLMA consistently outperforms PL-SVM and M3PL algorithms on both benchmark and CRC-MIPL datasets. This superiority can be attributed to two primary factors. First, within MIPLMA, the margin distribution loss simultaneously maximizes the mean margin and minimizes the variance of predicted probabilities. In contrast, PL-SVM and M3PL only maximize the mean margin of predicted probabilities. Therefore, our proposed margin distribution loss provides a more refined approach to optimizing the margins of predicted probabilities. Second, unlike the data degradation strategies employed by PLL algorithms for the indirect treatment of multi-instance bags, MIPLMA aggregates each multi-instance bag into a single bag-level feature representation using the margin-ware attention mechanism. The aggregated feature representations in MIPLMA are more discriminative than those obtained using data degradation strategies. Additionally, PL-SVM and M3PL employ iterative optimization strategies, rendering them challenging to integrate with deep models.

## C.2 Comparison with MIL Algorithms

Existing MIL algorithms are primarily tailored for addressing binary classification problems and are not readily applicable to solving MIPL problems. To overcome this limitation, we adopt the *One vs. Rest (OvR)* decomposition strategy employed in MIPLGP [16] for degradation of MIPL data. Specifically, when presented with a multi-instance bag $X_i$ linked to a candidate label set $S_i$, we assign each label from the candidate set to the bag, and thus generate $|S_i|$ multi-instance bags where each bag is associated with a singular bag-level label. For each class $c$ ($c \in \{1, 2, \cdots, k\}$), we train and test the $c$-th classifier by transforming the label $c$ to 1 (positive) and all other labels to 0 (negative). In the testing phase, a multi-instance bag yields $k$ predictions from the $k$ classifiers. If only one prediction is positive, the corresponding class label of that positive prediction is considered the classification result for the bag. In cases where there is more than one positive prediction among the $k$ predictions, we choose the class label associated with the classifier demonstrating the highest prediction confidence as the classification result for the bag. If all $k$ predictions are negative, the classification result is the class label with the lowest prediction confidence. In particular, LOSS-ATTEN is a multi-class MIL algorithm, eliminating the need for employing the One vs. Rest decomposition strategy. We can

Table A2: The classification accuracies (mean±std) of MIPLMA and comparative PLL algorithms with MLP on the benchmark datasets ($r \in \{1, 2, 3\}$).

| Algorithm | $r$ | MNIST-MIPL | | FMNIST-MIPL | | Birdsong-MIPL | | SIVAL-MIPL | |
|---|---|---|---|---|---|---|---|---|---|
| MIPLMA | 1 | **.985±.010** | | **.915±.016** | | **.776±.020** | | **.703±.026** | |
| | 2 | **.979±.014** | | **.867±.028** | | **.762±.015** | | **.668±.031** | |
| | 3 | **.749±.103** | | **.654±.055** | | **.746±.013** | | **.627±.024** | |
| | | Mean | MaxMin | Mean | MaxMin | Mean | MaxMin | Mean | MaxMin |
| PRODEN | 1 | .555±.033 | .465±.023 | .652±.033 | .358±.019 | .303±.016 | .339±.010 | .303±.020 | .322±.018 |
| | 2 | .372±.038 | .338±.031 | .463±.067 | .315±.023 | .287±.017 | .329±.016 | .274±.022 | .295±.021 |
| | 3 | .285±.032 | .260±.037 | .288±.039 | .265±.031 | .278±.006 | .305±.015 | .242±.009 | .244±.018 |
| RC | 1 | .660±.031 | .518±.022 | .697±.166 | .421±.016 | .329±.014 | .379±.014 | .344±.014 | .304±.015 |
| | 2 | .577±.039 | .462±.028 | .684±.029 | .363±.018 | .301±.014 | .359±.015 | .299±.015 | .268±.023 |
| | 3 | .362±.029 | .366±.039 | .414±.050 | .294±.053 | .288±.019 | .332±.024 | .256±.013 | .244±.014 |
| LWS | 1 | .605±.030 | .457±.028 | .702±.033 | .346±.033 | .344±.018 | .349±.013 | .346±.014 | .345±.013 |
| | 2 | .431±.024 | .351±.043 | .547±.040 | .323±.031 | .310±.014 | .336±.013 | .312±.015 | .314±.019 |
| | 3 | .335±.029 | .274±.037 | .411±.033 | .267±.034 | .289±.021 | .307±.016 | .286±.018 | .268±.019 |
| CAVL | 1 | .539±.048 | .497±.025 | .679±.031 | .359±.024 | .312±.014 | .332±.011 | .237±.010 | .220±.022 |
| | 2 | .380±.040 | .337±.030 | .482±.097 | .327±.021 | .285±.014 | .303±.022 | .197±.018 | .199±.014 |
| | 3 | .266±.021 | .267±.026 | .288±.060 | .266±.033 | .278±.014 | .282±.010 | .169±.017 | .144±.011 |
| POP | 1 | .505±.029 | .443±.032 | .641±.033 | .361±.030 | .319±.020 | .349±.015 | .376±.019 | .374±.014 |
| | 2 | .315±.022 | .281±.026 | .367±.037 | .292±.019 | .299±.020 | .337±.014 | .334±.029 | .344±.012 |
| | 3 | .258±.041 | .233±.026 | .256±.025 | .239±.018 | .281±.016 | .312±.022 | .302±.022 | .310±.022 |

Table A3: The classification accuracies (mean±std) of MIPLMA and comparative PLL algorithms with MLP on the real-world datasets.

| Algorithm | C-Row | | C-SBN | | C-KMeans | | C-SIFT | |
|---|---|---|---|---|---|---|---|---|
| MIPLMA | .444±.010 | | .526±.009 | | .557±.010 | | **.553±.009** | |
| | Mean | MaxMin | Mean | MaxMin | Mean | MaxMin | Mean | MaxMin |
| PRODEN | .405±.012 | **.453±.009** | .515±.010 | **.529±.010** | .512±.014 | **.563±.011** | .352±.015 | .294±.008 |
| RC | .290±.010 | .347±.013 | .394±.010 | .432±.008 | .304±.017 | .366±.010 | .248±.008 | .204±.008 |
| LWS | .360±.008 | .381±.011 | .440±.009 | .442±.009 | .422±.035 | .335±.049 | .338±.009 | .287±.009 |
| CAVL | .394±.014 | .376±.010 | .475±.017 | .356±.028 | .409±.009 | .483±.030 | .321±.014 | .279±.014 |
| POP | .399±.011 | .374±.019 | .513±.020 | .431±.026 | .503±.014 | .440±.017 | .333±.009 | .293±.021 |

directly assign all candidate labels of a multi-instance bag $X_i$ as the label for the multi-instance bag and obtain $|\mathcal{S}_i|$ multi-instance bags with a singular bag-level label.

Table A1 presents the classification accuracy of MIPLMA and seven comparative MIL algorithms. Across all scenarios, MIPLMA consistently demonstrates significantly superior performance compared to the seven MIL algorithms. Notably, among the compared MIL algorithms, LOSS-ATTEN consistently outperforms its counterparts, primarily due to its capability of directly addressing the multi-class MIL problem. While these algorithms yield reasonable results on the MNIST-MIPL and FMNIST-MIPL datasets, their performance diminishes severely on datasets with more intricate features, such as Birdsong-MIPL and SIVAL-MIPL. We attribute this challenge to the noise present in the labels due to data degradation, hindering the effective learning of these MIL algorithms.

## C.3 Results of PLL algorithms with MLP

Among the compared PLL algorithms, PRODEN[33], RC[34], LWS[35], and CAVL[11] can be utilized with either linear model or MLP. Tables 2 and 3 in the main body present the results obtained using the linear model. In this section, we present the results of MIPLMA and the comparative PLL algorithms with MLP on the benchmark and CRC-MIPL datasets in Tables A2 and A3, respectively.

Table A2 presents the performance comparison of MIPLMA with the four PLL algorithms utilizing the MLP on benchmark datasets. MIPLMA consistently outperforms the comparative PLL algorithms in all cases, demonstrating statistically significant superiority. Interestingly, when employing the MLP, the compared PLL algorithms do not consistently achieve superior outcomes compared to the linear model. This observation suggests that the linear model has already captured sufficient features for benchmark datasets, while employing the MLP may lead to overfitting.

Table A3 demonstrates that MIPLMA surpasses the comparative PLL algorithms utilizing the MLP in 37 out of 40 cases. Compared to the outcomes obtained using the linear classifiers, the results obtained using the MLP exhibit superior performance. Notably, on the complex CRC-MIPL-KMeans dataset, the enhancement with the MLP is more pronounced. This indicates that the linear model is

Table A4: Win/tie/loss counts of MIPLMA against the compared algorithms.

| | MIPLMA against | | | In total |
|---|---|---|---|---|
| | MIL algorithms | PLL algorithms | MIPL algorithms | |
| $r = 1$ | 28/0/0 | 104/0/0 | 9/2/1 | 141/2/1 |
| $r = 2$ | – | 104/0/0 | 10/1/1 | 114/1/1 |
| $r = 3$ | – | 104/0/0 | 7/4/1 | 111/4/1 |
| CRC-MIPL | – | 98/4/2 | 13/2/0 | 111/6/2 |
| **In total** | 28/0/0 | 410/4/2 | 39/9/3 | **477/13/5** |

inadequate for comprehensive feature learning on the CRC-MIPL dataset, while the MLP demonstrates a more comprehensive capability in feature learning.

## C.4 Win/tie/loss counts of Experimental Results

To ensure result reliability, we conduct pairwise t-tests with a significance level of $0.05$. Table A4 summarizes the win/tie/loss counts between MIPLMA and seven MIL, seven PLL, and three MIPL algorithms on benchmark datasets with varying false positive labels ($r \in \{1, 2, 3\}$), as well as the CRC-MIPL dataset. Several key insights can be drawn from our analysis: (a) MIPLMA shows statistical superiority over MIL, PLL, and MIPL in $100\%$, $98.6\%$, and $76.5\%$ of cases, respectively. (b) MIPLMA achieves statistical superiority in $97.3\%$ of cases on benchmark datasets. (c) For the CRC-MIPL dataset, MIPLMA shows statistical superiority in $91.6\%$ of cases. Overall, MIPLMA outperforms the compared algorithms in $96.4\%$ of cases.

# D Further Analyses

## D.1 Interpretability of the Attention Mechanism.

To investigate the interpretability of attention mechanisms, we compare MIPLMA with ELIMIPL and DEMIPL regarding the attention scores across three test multi-instance bags in the FMNIST-MIPL dataset ($r = 1$). Each row in Figure A2 represents a multi-instance bag, with the first to third columns depicting the attention scores produced by MIPLMA, ELIMIPL, and DEMIPL across the three multi-instance bags. It is noteworthy that since the sum of attention scores computed by DEMIPL does not equal 1, we initially normalize them to sum to 1 before visualization.

Figure A2 reveals several important findings. First, on the Bags 1 and 2, both our MIPLMA and ELIMIPL accurately identify positive instances. However, compared to ELIMIPL, MIPLMA assigns higher attention scores to positive instances and lower scores to negative ones, confirming the efficacy of our margin-aware attention mechanism. Second, the Bag 3 contains four positive instances, of which MIPLMA only identifies three, erroneously assigning high attention scores to several negative instances. However, the highest attention score assigned to a negative instance is still lower than the scores of these three positives. In contrast, ELIMIPL assigns significantly higher attention scores to two negative instances compared to the scores of the four positives, leading to a pronounced influence of negative instances on the aggregated bag-level features. Third, DEMIPL computes attention scores using the sigmoid function followed by normalization, while MIPLMA and ELIMIPL directly utilize the softmax function for attention score computation. Results indicate that attention scores computed by MIPLMA and ELIMIPL better conform to the distribution of positive and negative instances.

In summary, our margin-aware attention mechanism yields more accurate attention scores. Moreover, when distinguishing negative instances within multi-instance bags proves challenging, our attention mechanism effectively mitigates their influence on bag-level features.

## D.2 Comparison between Margin Loss and Margin Distribution Loss

To adjust the margin for the predicted probabilities in the label space, we first propose the margin loss $\lambda_{ml}$ as shown in Eq. (9). Whereas the margin loss aims to maximize the mean margin of predicted probabilities, the margin distribution loss endeavors to concurrently maximize this mean margin and minimize the variance in these probabilities. Therefore, the margin distribution loss is a generalized formulation of the margin loss. There are several significant connections between the two loss functions. (a) In scenarios where all $\phi_i$ are equal for $i \in \{1, 2, \cdots, m\}$, the variance

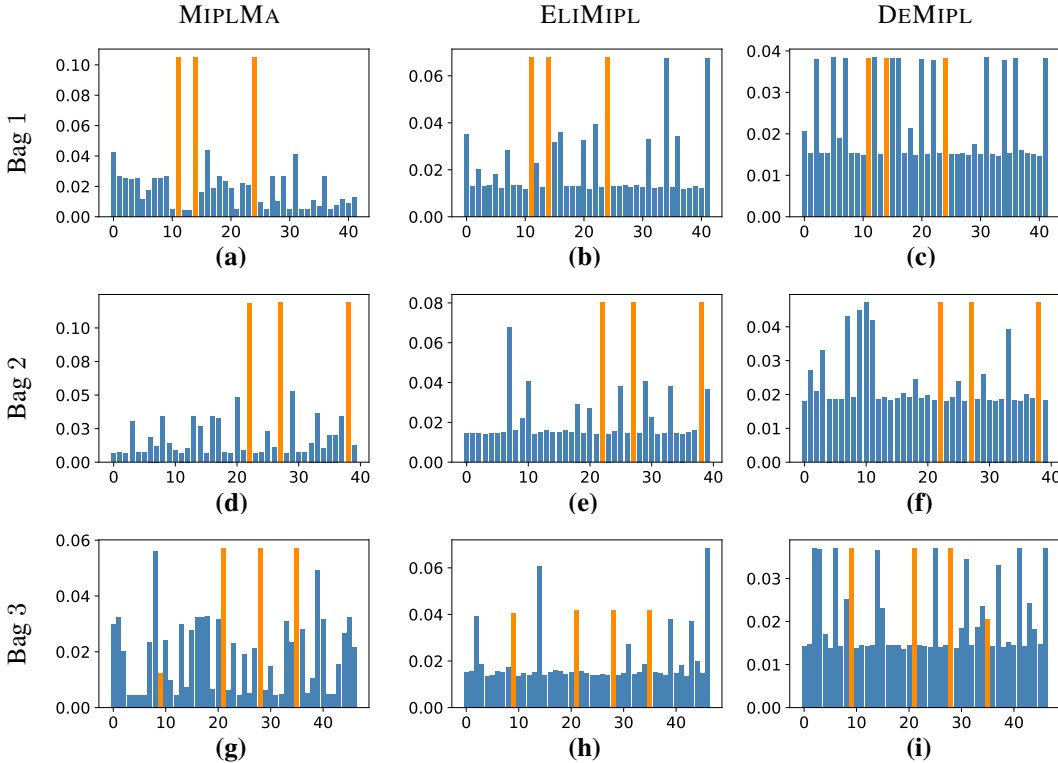

Figure A2: Attention scores of MIPLMA, ELIMIPL, and DEMIPL for three test bags in the FMNIST-MIPL dataset with $r = 1$. The horizontal axis denotes the indices of instances, while the vertical axis represents the corresponding attention scores. Orange and blue colors indicate attention scores assigned to positive and negative instances, respectively.

Table A5: The classification accuracies (mean±std) of MIPLMA and MIPL-MAMM on the benchmark datasets with the varying numbers of false positive labels ($r \in \{1, 2, 3\}$).

| Algorithm | $r$ | MNIST-MIPL | FMNIST-MIPL | Birdsong-MIPL | SIVAL-MIPL |
|---|---|---|---|---|---|
| MIPLMA | 1 | .985±.010 | **.915±.016** | **.776±.020** | **.703±.026** |
| | 2 | **.979±.014** | **.867±.028** | **.762±.015** | **.668±.031** |
| | 3 | .749±.103 | **.654±.055** | **.746±.013** | **.627±.024** |
| MIPL-MAMM | 1 | **.987±.008** | .894±.021 | .762±.015 | .682±.034 |
| | 2 | .969±.015 | .849±.025 | .741±.018 | .628±.017 |
| | 3 | **.756±.101** | .652±.061 | .671±.028 | .561±.031 |

$\mathcal{V}\{\cdot\}$ equals 0. Consequently, the margin distribution loss reduces to the margin loss. (b) When the variability of $\phi_i$ is low, as indicated by a small variance $\mathcal{V}\{\cdot\}$, the margin distribution loss slightly exceeds the mean of margin loss $\mathcal{M}\{\cdot\}$. This observation implies that when $\phi_i$ exhibits minimal variation across different samples, the margin distribution loss marginally surpasses the margin loss. (c) When the variability of $\phi_i$ is high, denoted by a large variance $\mathcal{V}\{\cdot\}$, the margin distribution loss $\mathcal{M}\{\cdot\}/(1 - \mathcal{V}\{\cdot\})$ can significantly exceed the mean of margin loss $\mathcal{M}\{\cdot\}$. In extreme cases, if $\mathcal{V}\{\cdot\}$ approaches 1, then the margin distribution loss becomes notably large. This suggests that when $\phi_i$ exhibits substantial variation across different samples, the margin distribution loss could notably surpass the margin loss.

To compare the magrin loss $\mathcal{L}_{ml}$ and the margin distribution loss $\mathcal{L}_m$, we substitute the $\mathcal{L}_m$ in Eq. (11) with $\mathcal{L}_{ml}$, resulting in a variant named MIPL-MAMM. Table A5 presents the classification accuracies of MIPLMA and MIPL-MAMM on the benchmark datasets, where the weight of the margin loss $\mathcal{L}_{ml}$ is tuned to achieve preferable performances. MIPLMA outperforms MIPL-MAMM in 10 of the 12 scenarios. On the MNIST-MIPL dataset, MIPLMA performs worse than MIPL-MAMM when $r = 1$ and 3. We speculate that this discrepancy may be attributed to the model slightly overfitting due to

the consideration of margin distribution on the relatively simple MNIST-MIPL dataset. Notably, the performance advantage of MIPLMA is more pronounced on the more challenging Birdsong-MIPL and SIVAL-MIPL datasets. This demonstrates that optimizing the margin distribution is more effective than only optimizing the margin mean, thus confirming the superiority of the margin distribution loss.

In summary, the margin distribution loss is associated with the variability of the margin loss across different samples, effectively indicating the concentration of the margin loss within a dataset. In the MIPL datasets, the margin distribution loss outperforms the margin loss in most cases.

### D.3 Robustness of the Weight for the Margin Distribution Loss

As shown in Eq. (11), $\lambda$ denotes the weighting factor for the margin distribution loss. Figure A3 depicts the average accuracies from ten runs. On the Birdsong-MIPL dataset, experimental findings encompass $\lambda$ values ranging from $\{2, 3, \cdots, 10\}$. The obtained accuracy remains relatively stable when $\lambda$ ranges between $3$ and $7$. However, when $\lambda$ exceeds $7$, accuracy decreases. Therefore, it is noteworthy that within a certain range of $\lambda$, MIPLMA can maintain a stable performance on the Birdsong-MIPL dataset. In the main experiment, $\lambda$ is set to $5$ for the Birdsong-MIPL dataset.

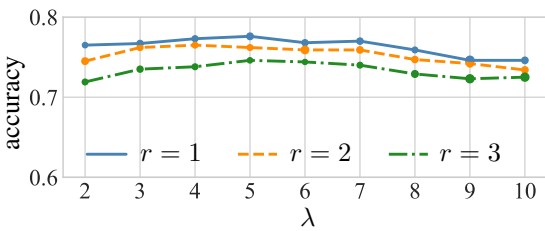

Figure A3: The classification accuracies (mean and std) of MIPLMA with varying $\lambda$ on Birdsong-MIPL dataset ($r \in \{1, 2, 3\}$). The diameter of the circle represents the relative standard deviation.

### D.4 Robustness of the Initial Temperature

Eq. (3) describes the annealing process of the temperature parameter $\tau^{(t)}$, with the initial temperature parameter $\tau^{(0)}$ manually specified. To investigate the influence of different initial temperature parameters on MIPLMA, we vary the initial temperature parameter across $\{4, 5, 6, 7, 8, 9, 10\}$, while maintaining all other parameters constant.

Figure A4 presents the average accuracies obtained from ten runs on the MNIST-MIPL (left) and Birdsong-MIPL (right) datasets. The experimental results suggest that the performance of MIPLMA remains relatively stable within the range of $\{4, 5, 6, 7, 8, 9, 10\}$, particularly noticeable when $r = 1$ and $r = 2$ on the MNIST-MIPL dataset. However, MIPLMA exhibits fluctuations in average accuracy when $r = 3$ on the MNIST-MIPL dataset, and the standard deviation is significantly higher compared to when $r = 1$ and $r = 2$. The MNIST-MIPL is a five-class dataset, and $r = 3$ represents an extremely challenging scenario. Therefore, we believe that the fluctuations in average accuracy observed in Figure A4 (left) are unavoidable. Furthermore, as shown in Figure 1, when $\tau^{(0)}$ varies within the range of $5$ to $10$, the performance of the Birdsong-MIPL dataset does not exhibit significant changes.

In summary, the results presented in Figure A4 highlight the robustness of MIPLMA to variations in the initial temperature parameter within the range of $\{5, 6, 7, 8, 9, 10\}$. In our experiments, the initial temperature parameter $\tau^{(0)}$ is consistently set to $5$ for all datasets.

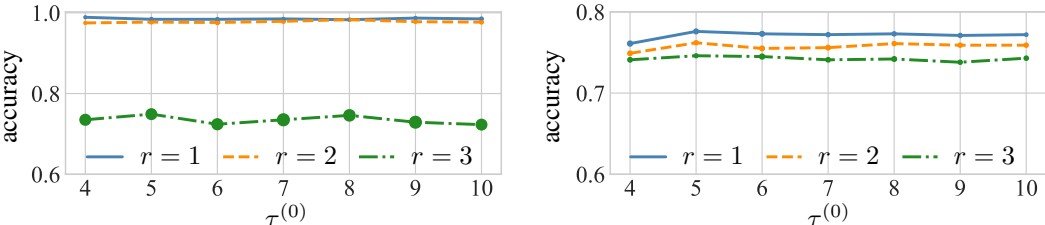

Figure A4: The classification accuracies (mean and std) of MIPLMA with varying $\tau^{(0)} \in \{4, 5, 6, 7, 8, 9, 10\}$ on MNIST-MIPL (left) and Birdsong-MIPL (right) datasets ($r \in \{1, 2, 3\}$). The diameter of the circle represents the relative standard deviation.

# E  Related Work

## E.1  Multi-Instance Learning

Multi-instance learning has its roots in drug activity prediction [45], and it has found applications in a variety of fields ranging from text classification [46, 42], object detection [47], and video anomaly detection [48]. In contemporary multi-instance learning methodologies, a prevalent strategy involves incorporating attention mechanisms to aggregate features from each multi-instance bag into a unified feature representation, subsequently fed into a classifier. Ilse et al. [4] introduced both the plain attention mechanism and gated attention mechanism to effectively enhance the performance of binary multi-instance learning. An extension of this paradigm is the loss-based attention mechanisms [43], providing a solution for multi-class tasks. Owing to the exceptional performance of the attention mechanisms, multi-instance learning methods based on attention mechanisms have gained widespread adoption in tasks such as histopathological image classification [49, 50]. These attention mechanisms typically fall under the category of soft attention mechanisms, wherein the weighted sum of attention scores for all instances in a multi-instance bag yields a bag-level feature representation. Conversely, Li et al. [51] introduced hard attention mechanisms, which focus on selecting a subset of instances from multi-instance bags to construct the feature representation.

Despite the considerable performance advancements achieved by these algorithms in multi-instance learning tasks, their direct application in multi-instance partial-label learning scenarios is impeded by their inability to handle inexact label information directly.

## E.2  Partial-Label Learning

Partial-label learning has widespread applications in diverse real-world scenarios, encompassing facial age estimation [37], face naming [9], object classification [52], and bioinformatics [29, 53]. Margin violations also exist in partial-label learning, prompting researchers to introduce the maximum margin criteria as a viable solution. Nguyen and Caruana [39] employed the maximum margin criterion to augment the distinction between the model's highest predicted probability on candidate labels and its highest predicted probability on non-candidate labels. Similarly, Yu and Zhang [38] focused on maximizing the margin between the model's predicted probability on the true label and its highest predicted probability on labels other than the true one. However, these methods require an alternating optimization, contributing to the intricacies of the optimization procedure. In recent years, numerous deep learning-based partial-label learning algorithms have emerged. Lv et al. [33] utilize linear classifiers or multi-layer perceptrons to learn feature representations from instances, employing progressive disambiguation strategies to identify true labels. Following this line of thought, Feng et al. [34] delved into the generation process of partial-label data and proposed two theoretically guaranteed partial-label learning algorithms. Similarly, Wen et al. [35] introduced a weighted loss for disambiguation, serving as a generalized version across multiple algorithms.

While these algorithms exhibit considerable efficacy in tackling partial-label learning problems, they encounter limitations in directly handling inexact supervision within the instance space. Consequently, they cannot be directly applied to multi-instance partial-label learning problems.

## E.3  Multi-Instance Partial-Label Learning

MIPL is an extension that encompasses both MIL and PLL. Its objective is to tackle the challenge of inexact supervision simultaneously present in both instance and label spaces. To our knowledge, only three viable MIPL algorithms, i.e., MIPLGP [16], DEMIPL [21], and ELIMIPL [22], currently exist. Tang et al. [16] have introduced the MIPL framework, with MIPLGP adopting an instance-space paradigm. MIPLGP is structured in three steps. Firstly, it augments a negative class for each candidate label set. Secondly, it treats the candidate label set of each multi-instance bag as that of each instance within the bag. Finally, it employs the Dirichlet disambiguation strategy and the Gaussian processes regression model for disambiguation. On the other hand, DEMIPL follows the embedded-space paradigm and consists of two steps. Initially, it aggregates each multi-instance bag into a unified feature representation through the disambiguated attention mechanism. Subsequently, it employs a momentum-based disambiguation strategy to discern true labels from candidate label sets. Following this way, Tang et al. [22] have proposed ELIMIPL to exploit the information from candidate and non-candidate label sets via three loss functions. Specifically, ELIMIPL learns the mappings from

Table A6: Code availability of the algorithms.

| Algorithm | URL |
| --- | --- |
| MIPLMA | https://github.com/tangw-seu/MIPLMA |
| ELIMIPL | https://github.com/tangw-seu/ELIMIPL |
| DEMIPL | https://github.com/tangw-seu/DEMIPL |
| MIPLGP | https://github.com/tangw-seu/MIPLGP |
| VWSGP | https://github.com/melihkandemir/vwsgp |
| VGPMIL | https://github.com/manuelhaussmann/vgpmil |
| LM-VGPMIL | https://github.com/manuelhaussmann/vgpmil |
| MIVAE | https://github.com/WeijiaZhang24/MIVAE |
| ATTEN | https://github.com/AMLab-Amsterdam/AttentionDeepMIL |
| ATTEN-GATE | https://github.com/AMLab-Amsterdam/AttentionDeepMIL |
| LOSS-ATTEN | https://github.com/xsshi2015/Loss-Attention |
| PRODEN | https://github.com/Lvcrezia77/PRODEN |
| RC | https://lfeng-ntu.github.io/codedata.html |
| LWS | https://github.com/hongwei-wen/LW-loss-for-partial-label |
| CAVL | https://github.com/Ferenas/CAVL |
| POP | https://github.com/palm-ml/POP |
| PL-AGGD | http://palm.seu.edu.cn/zhangml/ |

the multi-instance bags to the candidate label sets and the sparsity of the candidate label matrix. Moreover, it incorporates the non-candidate label information via an inhibition loss. Recently, Yang et al. [54] have proposed a probabilistic generative model for multi-instance partial-label learning (MIPL) that infers latent ground-truth labels by modeling the data generation process of MIPL. From a theoretical perspective, Wang et al. [55] have established connections between MIPL and latent structural learning, as well as neurosymbolic integration.

However, the existing MIPL algorithms do not consider the margins of attention score and predicted probabilities, and thus suffer from the issues of margin violations illustrated in Figure 1.

## F    Data and Code Availability

The implementations of the compared algorithms are publicly available. Table A6 includes the URLs of all compared algorithms in this paper, while the source code of our proposed MIPLMA is included in the supplementary material. The MIPL datasets can be accessed publicly at http://palm.seu.edu.cn/zhangml/.

## G    Broader Impact

The proposed MIPLMA has several potential societal impacts, both positive and negative.

The positive impacts of MIPLMA include: (a) Medical diagnosis: MIPLMA could enhance the accuracy of medical diagnoses where obtaining exact labels is challenging. For example, in medical image classification tasks, when experts lack confidence in the provided results, MIPLMA can yield more accurate outcomes. (b) Privacy-preserving surveys: Similar to PLL methods, MIPLMA can be applied in scenarios where respondents are hesitant to disclose sensitive information. By allowing respondents to select a set of candidate labels instead of providing a single label, MIPLMA can facilitate data collection while respecting privacy. This can be particularly useful in fields such as mental health, where patients may feel uncomfortable disclosing exact symptoms or conditions.

Conversely, there may also exist some negative impacts of MIPLMA. (a) Misuse in surveillance: There is a risk that our method could be exploited in surveillance systems to infer sensitive information about individuals without their explicit consent. For example, an adversary could use our algorithm to analyze multi-instance data collected from social media or other sources to infer private details about individuals, leading to potential breaches of privacy. (b) Job displacement: As MIPLMA improves the efficiency and effectiveness of learning from inexact data, it might reduce the demand for human annotators. This could lead to job displacement involved in data labeling and annotation. Consequently, efforts should be made to retrain and upskill these workers to mitigate the impact.

