# OpenReview forum: "Multi-Instance Partial-Label Learning with Margin Adjustment"
_NeurIPS.cc/2024/Conference — NeurIPS 2024 poster_

### Official Review · Reviewer_Unv6 · 2024-07-07

**Soundness:** 3
**Presentation:** 3
**Contribution:** 3
**Rating:** 7
**Confidence:** 4

**Summary:**

The paper proposes an approach to tackle the multi-instance partial-label learning (MIPL) problem. In MIPL, each training sample is represented as a bag of instances associated with a set of candidate labels. The existing MIPL algorithms may gives the high prediction probability on non-candidate label, which can lead to suboptimal performance. They propose MIPLMA  algorithm addresses this issue by introducing a margin-aware attention mechanism and a margin-compliant loss function. The algorithm dynamically adjusts the margins for attention scores, while the loss function ensures that the margins between predicted probabilities on candidate and non-candidate label sets are properly constrained. The paper presents experimental results on benchmark and real-world datasets that demonstrate the superior performance of MIPLMA compared to existing MIPL algorithms, as well as other multi-instance algorithms and partial-label learning algorithms.

**Strengths:**

1. The paper is well-organized and easy to comprehend.
2. The experiments are generally comprehensive.
3. Theoretical analysis contributes to establishing the effectiveness of the proposed model.

**Weaknesses:**

1. The margin-aware attention is performed by setting a changing temperature parameter which seems a trival idea. And the details of how the gap of attention scores help the model to decrease the prediction probability on non-candidate labels are also not given.
2. The form of margin-compliant loss seems not easy to be optimized in neural network framework, the paper should give more details on the proposed loss function.

**Questions:**

1. The initial weights are uniformly distributed. After the training in first epoch, the weights change. Subsequent epochs can only make the margin of the distribution larger, so the results are greatly affected by the first epoch. If there are errors in the training of the first few epochs, Will this error accumulate?

**Limitations:**

Yes.

---

> ### Author Rebuttal · Authors · 2024-08-07
>
> Thank you for your thorough review and positive feedback on our paper. We are pleased that you found the paper well-organized, comprehensive, and theoretically supported. Below, we have summarized your comments and provided our responses accordingly.
>
> >  The details of how the gap of attention scores help the model to decrease the prediction probability on non-candidate labels are also not given.
>
> Our paper utilizes margin-compliant loss to increase the gap between prediction probabilities for candidate and non-candidate labels, thus reducing the probability for non-candidate labels. To examine the impact of attention score gaps, we introduce MIPLMA-NOTEM, a variant of MIPLMA without the temperature parameter in the attention mechanism.
>
> In experiments with the MNIST-MIPL dataset ($r=1$), both models yield similar prediction probabilities of approximately 0.088 for non-candidate labels, showing no significant difference. Visualization of attention scores for three multi-instance bags from the training set, shown in `Figure R2` of the attached PDF, where the horizontal axis denotes the indices of instances, while the vertical axis represents the corresponding attention scores. Red and blue colors indicate attention scores assigned to positive and negative instances, respectively. The results reveals that the temperature parameter effectively increases the gap between attention scores for positive and negative instances.
>
> Furthermore, `Figure R3` of the attached PDF compares feature visualizations on the test set. MIPLMA produces well-clustered features by class, while MIPLMA-NOTEM exhibits errors, such as misclassifications where points from the first class incorrectly approach clusters of other classes. Therefore, while the temperature parameter may not directly impact prediction probabilities for non-candidate labels, it enhances feature representation accuracy. We will include these findings and discussions in the revised manuscript.
>
>
>
> > The margin-compliant loss seems not easy to be optimized in neural network framework. More details are required.
>
> The core of the margin-compliant loss involves finding the maximum predicted probabilities for the candidate and non-candidate label sets, concatenating the margins, and calculating their mean and standard deviation. First, we use `torch.max` to obtain the maximum predicted probabilities for both sets. Then, we employ `torch.cat` to concatenate the margins in the current and previous batches. Finally, we calculate the mean and standard deviation of the margins using `.mean` and `.std` operations, followed by the value of the margin-compliant loss. All these operations, including `torch.max`, are differentiable in PyTorch. We will provide additional details on the margin-compliant loss in the revised manuscript to clarify its optimization within the neural network framework.
>
>
>
> > Will errors in the first few epochs affect subsequent training?
>
> In MIPL, where prior knowledge of true labels in the candidate label sets is unavailable, initializing candidate label weights with averages is reasonable. Although early errors might accumulate and impact later training, our dynamic disambiguation loss reduces the influence of early predictions by assigning them lower weights, as shown in Eq. (8). To further investigate the effectiveness of our dynamic disambiguation loss, we also introduce a variant without the dynamic disambiguation, MIPLMA-$\alpha$, where $\alpha^{(t)}$ is set to 0 for all $t \in \\{1,2,\cdots,T\\}$.
>
> The table below shows the performance of MIPLMA and MIPLMA-$\alpha$ on benchmark datasets. MIPLMA-$\alpha$ performs well on simpler disambiguation tasks (e.g., $r=1$ or $r=2$) but shows decreased performance compared to MIPLMA as the number of false positive labels increases (e.g., $r=3$). The gap between MIPLMA and MIPLMA-$\alpha$ widens with greater disambiguation challenges. These results suggest that our dynamic disambiguation loss is effective iin mitigating the errors in the first few epochs.
>
> | Algorithm       | r    | MNIST-MIPL | FMNIST-MIPL | Birdsong-MIPL | SIVAL-MIPL |
> | --------------- | ---- | :--------: | :---------: | :-----------: | :--------: |
> |                 | 1    | .985±.010  |  .915±.016  |   .776±.020   | .703±.026  |
> | MIPLMA          | 2    | .979±.014  |  .867±.028  |   .762±.015   | .668±.031  |
> |                 | 3    | .749±.103  |  .654±.055  |   .746±.013   | .627±.024  |
> |                 | 1    | .991±.025  |  .897±.022  |   .772±.020   | .704±.022  |
> | MIPLMA-$\alpha$ | 2    | .949±.056  |  .841±.021  |   .765±.027   | .658±.021  |
> |                 | 3    | .661±.161  |  .538±.053  |   .721±.042   | .586±.026  |

---

> > ### Author Response · Authors · 2024-08-12
> > **Looking Forward to Your Feedback**
> >
> > Dear Reviewer Unv6,
> >
> > Thank you for your valuable feedback, which has greatly improved our paper. We have carefully addressed your concerns by providing visualizations of attention scores and bag-level features, detailing the optimization of the margin-compliant loss, and demonstrating how our dynamic disambiguation loss mitigates early errors. Should any issues remain, we are happy to discuss them further.
> >
> > Best regards,
> >
> > The Authors

---

### Official Review · Reviewer_b79F · 2024-07-09

**Soundness:** 3
**Presentation:** 4
**Contribution:** 4
**Rating:** 7
**Confidence:** 5

**Summary:**

In this paper, the authors observe the presence of ‘margin-violations’ in the MIPL problem, which manifest in two aspects: between positive and negative instances, and between candidate and non-candidate labels. Therefore, the paper proposes modifications to the attention mechanism and enhancement of the loss function to rectify and widen the margin.

**Strengths:**

1. This paper aims to address a challenging scenario, specifically the MIPL problem, and identifies an intriguing phenomenon known as ‘margin violations’.
2. This work proposes MIPLMA to solve this issue, which combines several effective but not complicated tricks.
3. The authors conduct extensive experiments and compare with a number of methods to valid the effectiveness of the proposal.

**Weaknesses:**

1. In Figure 1, the authors compare the performances of MIPLMA and DEMIPL, however, according to the results in Table 2 and Table 3, ELIMIPL surpasses DEMIPL on FMNIST-MIPL, and both ELIMIPL and MIPLGP perform better than DEMIPL on CRC-MIPL-Row dataset, so the comparison with these two methods would be more beneficial.
2. When discussing the linear-decay strategy for temperature in Equation 3, the sudden introduction of the constant coefficient 0.95 requires some clarification for better understanding.
3. In Sec.3.1.1, the authors claim that they are the first to employ ResNet on CRC-MIPL dataset, so it is suggested that the authors should list the detailed baseline performances on C-R34-16 and C-R34-25 datatset bag generators, just as in Table.3.
4. The authors assert that “features learned by ResNet-34 outperform those obtained via image bag generators in terms of classification performance.” However, it remains unclear whether this still holds on other benchmarks such as FMNIST-MIPL. To strengthen this claim, it would be beneficial to add further experiments or discussions on this.

**Questions:**

See weaknesses

---

> ### Author Rebuttal · Authors · 2024-08-07
>
> Thanks for your detailed and positive feedback on our paper. We are glad you found our approach to addressing ‘margin violations’ in the MIPL problem intriguing. Below, we address your comments and questions.
>
> > In Figure 1, a comparison with ELIMIPL and MIPLGP would be more beneficial.
>
> MIPLGP follows the instance-space paradigm, which assigns augmented candidate label sets of bags to each instance and aggregates bag-level labels from instance-level labels. As a result, we cannot observe the phenomenon of margin violations in the instance and label spaces. For ELIMIPL, we will update Figure 1 to include its results in the revised manuscript.
>
>
>
> >  The constant coefficient 0.95 in Equation 3 requires clarification.
>
> The decay rate of $0.95$ for the temperature parameter in Equation 3 was selected based on preliminary experiments, which indicated that the performance of the margin-aware attention mechanism is relatively stable across a range of decay rates. To maintain consistency, we fixed the decay rate at $0.95$ for all datasets. We have conducted additional experiments on the FMNIST-MIPL dataset with decay rates from $0.9$ to $0.99$, as shown in `Figure R1` of the attached PDF. These experiments revealed classification accuracies between $0.907$ and $0.927$. We will provide these details and results in the revised manuscript.
>
>
>
> > List detailed baseline performances on C-R34-16 and C-R34-25 datasets, as in Table 3.
>
> Thank you for your suggestion. The following table shows the classification accuracies (mean±std) for MIPLMA, ELIMIPL, and DEMIPL on the C-R34-16 and C-R34-25 datasets. MIPLMA shows a slight improvement over ELIMIPL and DEMIPL on the C-R34-16 dataset, and a more significant advantage on the C-R34-25 dataset. We will include more baseline performances in Table 3 of the revised manuscript.
>
> | Algorithm | C-R34-16  | C-R34-25  |
> | --------- | :-------: | :-------: |
> | MIPLMA    | .631±.008 | .685±.011 |
> | ELIMIPL   | .628±.009 | .663±.009 |
> | DEMIPL    | .625±.008 | .650±.010 |
>
>
>
> > It remains unclear whether the superior performance of features learned by ResNet-34 holds on other benchmarks such as FMNIST-MIPL.
>
> The instance-level features of the Birdsong-MIPL and SIVAL-MIPL datasets are preprocessed, which prevents the use of DCNNs like ResNet-34 for feature learning. We have tested LeNet and ResNet-34 on the MNIST-MIPL and FMNIST-MIPL datasets, but the classification accuracies were unsatisfactory. We believe this is due to the relatively simple nature of the features in these datasets, which are adequately captured by simple networks like the two-layer CNN used in our study. While deep convolutional neural networks might perform better on fully supervised MNIST and FMNIST datasets, their benefits may be less evident in weakly supervised scenarios.

---

> > ### Author Response · Authors · 2024-08-12
> > **Looking Forward to Your Feedback**
> >
> > Dear Reviewer b79F,
> >
> > Thank you for your constructive feedback, which has significantly enhanced our manuscript. We have addressed your concerns by conducting experiments on the temperature parameter decay rate, providing results for the C-R34-16 and C-R34-25 datasets, and offering further explanations on the feature extractor. If there are any remaining issues, we are open to further discussion.
> >
> > Best regards,
> >
> > The Authors

---

### Official Review · Reviewer_BX8M · 2024-07-14

**Soundness:** 2
**Presentation:** 2
**Contribution:** 2
**Rating:** 5
**Confidence:** 4

**Summary:**

This paper deals with an emerging learning framework, i.e., multi-instance partial-label learning framework, which can be regarded as an extension and combination of multi-instance learning and partial-label learning. Overall, such dual inexact supervision makes the MIPLL difficult to resolve. This paper introduces a margin-aware attention mechanism and margin-compliant loss to deal with such issue.

**Strengths:**

1. The motivation is clear and straightforward;
2. The paper is well-written and easy-to-follow.
3. Empirical results show satisfactory performance on several benchmarks as compared with existing methods.

**Weaknesses:**

1. I am confused with the definition of MIPLL as the authors claimed in the paper: "positive instances refer to the instances that belong to the true label in the setting of MIPL"，I agree with the definition of positive instances. However, for the negative instances, " while negative instances represent the remaining instances in the bag that are not associated with any label in the label space". Does the author indicates that the negative instances are not associated with the whole label space or just the candidate label set for the given bag?  I think the authors should be careful with this definition. I don't think the whole label space is the correct description w.r.t the negative instances.
2. The so called margin-aware attention mechanism and the margin-compliant loss are not novel, which in my opinion have already been utilized in other well-defined issues. So the novelty cannot reach the bar of NeruIPS.
3. In addition, although the multi-instance partial label learning framework is relatively novel and interesting, there has already existed at least two pioneer works, which again decreases the novelty of this work.
4. As compared with existing methods, the proposed algorithm cannot achieve the dominant performance on all the datasets under all the settings.

**Questions:**

N/A

**Limitations:**

See the weakness.

---

> ### Author Rebuttal · Authors · 2024-08-07
>
> Thank you for your detailed review and constructive feedback. In the following, we address your comments and concerns.
>
> > Definition of negative instances.
>
> In line with previous MIPL work such as MIPLGP, DEMIPL, and ELIMIPL, we do not associate negative instances with the label space. For the MNIST-MIPL dataset, positive instances are drawn from the target classes $\\{0, 2, 4, 6, 8\\}$, while negative instances come from the reserved classes $\\{1, 3, 5, 7, 9\\}$. Positive instances in a multi-instance bag are sampled from target classes, and negative instances are sampled from reserved classes. Therefore, the true label of a multi-instance bag corresponds to the actual label, with candidate labels sampled from the remaining target classes.
>
> Let us consider two scenarios:
>
> 1. Negative instances not associated with the labels space: If a test multi-instance bag has positive instances from class $0$ and negative instances from $\\{1, 3, 5, 7, 9\\}$, correct classification requires the classifier to predict class $0$.
> 2. Negative instances associated with the label space: If a test multi-instance bag has positive instances from $\\{0, 2\\}$ and negative instances from $\\{1, 3, 5, 7, 9\\}$, predicting either class $0$ or class $2$ is not considered wrong. However, this conflicts with the multi-class classification principle, which assumes each sample belongs to a single class.
>
> Not associating negative instances with the label space is reasonable for MIPL applications. For example, For the CRC-MIPL dataset, positive instances are cell types like lymphocytes and colorectal adenocarcinoma epithelium, while negative instances refer to background information, such as non-cellular areas or areas without significant tissue. If background is included in the label space, the true label for each multi-instance bag should reflect both the background and the cell type class, conflicting with the multi-class classification principle that each sample should belong to a single class.
>
>
>
> >  The margin-aware attention mechanism and margin-compliant loss are not novel.
>
> We recognize that the attention mechanism and margin-based loss have been used in MIL and PLL, respectively. However, our paper introduces a novel perspective by identifying margin violations in both the instance and label spaces with a dual margin adjustment strategy. Existing margin-based PLL methods, such as PL-SVM and M3PL, have two main drawbacks: they rely on iterative optimization, making integration with neural networks difficult, and they only maximize the mean margin of predicted probabilities. In contrast, our margin-compliant loss integrates seamlessly with neural networks and both maximizes the mean margin and minimizes the standard deviation of predicted probabilities.
>
> While the attention mechanism and the margin-based methods are not novel in MIL or PLL, their effective integration and enhancement within the MIPL framework represent a significant contribution. We will elaborate on these unique aspects and improvements in the revised manuscript.
>
>
>
> > Existing works decrease the novelty of this work.
>
> Existing works, such as MIPLGP, DEMIPL, and ELIMIPL, provide important prior contributions to the MIPL framework. These can be divided into two categories: (1) MIPLGP uses a Gaussian Process Regression model for fitting transformed MIPL data, but it is sensitive to negative instances and requires more computational resources; (2) DEMIPL and ELIMIPL leverage attention mechanisms to learn global feature representations, yet they encounter margin violations in both instance and label spaces.
>
> Our work distinguishes itself by being the first to identify and address margin violations in MIPL. We propose an effective dual margin adjustment strategy to counter these issues. Additionally, we introduce CRC-MIPL datasets with deep feature extractors, specifically the C-R34-16 and C-R34-25 datasets, which is also a novel contribution in MIPL.
>
>
>
> > The proposed algorithm does not achieve dominant performance on all the datasets under all the settings.
>
> In machine learning, the "No Free Lunch" theorem states that no algorithm can outperform all others across every task and data distribution. Table A4 in the Appendix presents the results of pairwise t-tests at a significance level of 0.05 between our proposed MIPLMA and the comparison algorithms. Out of $430$ comparisons, MIPLMA outperforms the comparison algorithms in $412$ cases and shows no significant difference in $15$ cases. Thus, while MIPLMA may not dominate in every scenario, it demonstrates substantial superiority in the majority of cases.

---

> > ### Author Response · Authors · 2024-08-12
> > **Looking Forward to Your Feedback**
> >
> > Dear Reviewer BX8M,
> >
> > Thank you for your valuable feedback, which has significantly improved our paper. We have carefully addressed your concerns regarding the definition of negative instances and the novelty of our work. We have reviewed and discussed two related MIPL studies [A, B] with different negative instance setups, which will be discussed in the revised manuscript. Should any issues remain, we welcome further discussion.
> >
> > Best regards,
> >
> > The Authors
> >
> >
> >
> > [A] Wang et al. On learning latent models with multi-instance weak supervision. NeurIPS 2023.
> >
> > [B] Wang et al. On characterizing and mitigating imbalances in multi-instance partial label learning. arXiv:2407.

---

> > ### Comment · Reviewer_BX8M · 2024-08-14
> >
> > Thanks for the author's rebuttal, and most of my concerns have been fixed. So I would like to raise my score.

---

> > > ### Author Response · Authors · 2024-08-14
> > >
> > > Thank you for your positive feedback!

---

### Official Review · Reviewer_zb3e · 2024-07-16

**Soundness:** 3
**Presentation:** 3
**Contribution:** 3
**Rating:** 7
**Confidence:** 4

**Summary:**

In this paper, the learning scenario of multi-instance partial learning (MIPL) is studied. The paper proposes a new approach for MIPL, whose key technical idea is margin regularization. The paper points out that an important issue for previous MIPL approaches is the ignorance of margin information in both the outputs of attention modules and final predictors, leading to incorrect attention weights and prediction confidence. To address this issue, the paper proposes a novel margin loss for MIPL, whose effectiveness is verified under various MIPL benchmark datasets.

**Strengths:**

1. In my view, the idea of margin regularization for MIPL is quite interesting, in special the regularization for the margins of attention weights. Even though attention weights are usually considered as a reflection of feature correlations, the accuracy of this correlation is difficult to quantify and receives less study. I think the margin measure is a good point for this quantification, and this idea can be indeed useful in the MIPL problem.

2. The proposed method revives the technique of margin distribution optimization. I like this idea due to its elegance in quantifying margin information. This also leads to the future possibility of theoretical studies for MIPL since the margin distribution framework is well theoretical grounded.

3. The experimental results are fruitful. The performance gain over baseline methods is significant.

**Weaknesses:**

1. The discussion on the scheme of modifying the strength of margin regularization can be augmented.

**Questions:**

I suggest including more discussions on setting the parameter \lambda. In my view, this parameter could significantly affect the learning performance. In my understanding, this parameter can be kept small at the beginning and be gradually increased. The reason is discussed in the paper: at the beginning, the model is less accurate, making the margin less informative, and the thing turns different as learning process proceeds. I find that currently, \lambda is kept as a constant. So I would like to ask for more discussions on the interpretations for this choice.

**Limitations:**

Yes

---

> ### Author Rebuttal · Authors · 2024-08-07
>
> Thank you for your insightful comments and positive feedback on our paper. We are glad you found the idea of margin regularization and our results valuable. Below, we answer your questions.
>
> > More discussion on setting the parameter $\lambda$. The parameter could significantly affect learning performance and might be better if varied during training rather than kept constant.
>
> To address the concern about the parameter $\lambda$, we conducted further study using a dynamic adjustment strategy defined as $\lambda(t) = \min\\{\frac{t}{T^\prime} \lambda^\prime, \lambda^\prime\\}$, where $t$ denotes the current epoch, and $T^\prime$ and $\lambda^\prime$ control the rate of increase and the maximum value of $\lambda$, respectively. We name this model MIPLMA-$\lambda$. When $T^\prime=1$, MIPLMA-$\lambda$ is equivalent to MIPLMA. We conducted experiments on the MNIST-MIPL dataset with $T^\prime$ set to $\\{1,10,50,100\\}$, using the same $\lambda^\prime$ as in MIPLMA. Results show that dynamically adjusting $\lambda$ can sometimes improve classification accuracy compared to a constant $\lambda$; however, in other cases it may perform worse. Notably, when $r=3$, MIPLMA-$\lambda$ tends to enhance accuracy more significantly.
>
> We sincerely appreciate your suggestion for dynamic adjustment of $\lambda$. Designing effective adjustment strategies is indeed a valuable research direction. We will explore more refined dynamic adjustment methods for $\lambda$ in the future.
>
> | Algorithm        | $r$  | $T^\prime=1$ | $T^\prime=10$ | $T^\prime=50$ | $T^\prime=90$ |
> | ---------------- | :--: | :----------: | :-----------: | :-----------: | :-----------: |
> |                  | $1$  |  .985±.010   |   .957±.056   |   .986±.007   |   .987±.009   |
> | MIPLMA-$\lambda$ | $2$  |  .979±.014   |   .970±.015   |   .979±.011   |   .977±.014   |
> |                  | $3$  |  .749±.103   |   .763±.089   |   .753±.101   |   .738±.105   |

---

> ### Author Response · Authors · 2024-08-12
> **Looking Forward to Your Feedback**
>
> Dear Reviewer zb3e,
>
> Thank you for your insightful feedback, which has greatly enhanced our paper. We have carefully addressed your concerns, including conducting experiments on the dynamic adjustment strategy for the parameter $\lambda$. If you have any further questions or concerns, we would be pleased to discuss them.
>
> Best regards,
>
> The Authors

---

### Official Review · Reviewer_H7jM · 2024-07-23

**Soundness:** 3
**Presentation:** 3
**Contribution:** 2
**Rating:** 5
**Confidence:** 4

**Summary:**

The paper proposes MIPLMA, a new Multi-Instance Partial Label algorithm that focuses on margin adjustments in both instance and label space. While computing the bag level representations, it introduces a temperature parameter in the margin-aware attention mechanism to widen the gap between attention scores for positive and negative instances. It also employs a margin loss to maximise the margin between the highest predicted probability on the candidate label set and on the non-candidate label set.

**Strengths:**

- Introduces and adapts margin-based approaches from weakly supervised learning paradigms for multi-Instance PLL

- The paper is well-written, and I was able to understand it clearly

**Weaknesses:**

- In many cases, either the number of classes is low or the number of false positives is low, which raises concerns regarding the generalization of the algorithm

- Paper has not been compared with some of the recent PLL algorithms like PICO [A], etc.  A discussion on why such comparisons were not done would be helpful. Maybe adaptation was difficult, or they focused on a single modality?

- Contributions are limited, it is a direct adaptation of margin-based losses to the problem

- The applications of this setup need to be more convincing to me. They do give one application on a Cancer detection dataset, however, I believe it could be achieved through an architecture capable of processing higher-resolution images.

[A] Wang et al. PiCO: Contrastive Label Disambiguation for Partial Label Learning. ICLR 2022

**Questions:**

- The creation of the synthetic datasets is unclear. While Table 1 presents details regarding the number of instances in a bag, the distribution proportion of true positives in a bag is not mentioned. This could have implications for the performance of the algorithm.


- In section 3.1.2 (line 200), It is mentioned that “PLL algorithms can be equipped with either the linear model or multi-layer perceptrons (MLP) as backbone networks”. However, DNNs like ResNet can be used to learn feature representations. Ideally, the comparison should be made using the same feature extractors, the ones mentioned in lines 208 - 213.

- I would willing to reconsider my ratings post the author rebuttal

**Limitations:**

Yes

---

> ### Author Rebuttal · Authors · 2024-08-07
>
> Thank you for your detailed review and valuable comments aimed at improving our paper. Below, we provide a summary of your comments along with our corresponding responses.
>
> > Concerns regarding generalization due to low class numbers or false positives.
>
> We conducted additional experiments on the SIVAL-MIPL dataset with higher number of false-positive labels ($r \in \\{4,5,6,7,8,9\\}$). The results in the following table show that MIPLMA achieves superior performance in all cases, which demonstrate its robustness to various number of false positive labels.
>
> | Algorithm |   $r=4$   |   $r=5$   |   $r=6$   |   $r=7$   |   $r=8$   |   $r=9$   |
> | --------- | :-------: | :-------: | :-------: | :-------: | :-------: | :-------: |
> | MIPLMA    | .618±.012 | .584±.019 | .524±.029 | .510±.030 | .431±.026 | .371±.051 |
> | ELIMIPL   | .559±.038 | .507±.022 | .439±.025 | .392±.031 | .343±.029 | .339±.030 |
> | DEMIPL    | .470±.036 | .413±.020 | .352±.032 | .323±.029 | .266±.022 | .275±.021 |
>
>
>
> > Lack of comparison with recent PLL algorithms like PiCO.
>
> We did not compare our method with PiCO for two main reasons:
>
> - PiCO relies on data augmentation to learn diverse features of partially labeled images. However, the features of the Birdsong-MIPL, SIVAL-MIPL, and CRC-MIPL datasets are preprocessed tabular data. This makes it challenging to apply data augmentation-based PLL methods to these datasets.
> - The instances in the MNIST-MIPL, FMNIST-MIPL, C-R34-16, and C-R34-25 datasets are raw images, which allows data augmentation-based PLL methods to learn features for each instance. However, these methods cannot aggregate a bag of instances into a unified feature representation as they lack an attention mechanism or similar aggregation method.
>
> In our submission, we compared our method with five PLL methods published between 2020 and 2022. We have now included an additional comparison with a PLL method called POP (ICML 2023). As shown in `Table R1` and `R2`  in the attached PDF, the main results indicate that while POP performs better than other PLL methods in most cases, it is inferior to our MIPLMA. We will include the above discussion and the results of POP in the revised manuscript.
>
>
>
> > A direct adaptation of margin-based losses.
>
> We acknowledge that margin-based losses have been studied in existing methods in PLL such as PL-SVM and M3PL. However, these methods have two drawbacks that make them unsuitable for the MIPL problem: they rely on iterative optimization strategies, making them difficult to integrate with neural networks; furthermore, they only maximize the mean margin of the predicted probabilities. Our proposed margin-compliant loss integrates naturally easily with neural networks and simultaneously maximizes the mean margin while minimizing the variance of predicted probabilities.
>
> To further illustrate this, we propose a variant of MIPLMA named MIPLML, which directly utilizes the margin loss $\mathcal{L}\_{\text{ml}}$ (Eq. 9). Specifically, the loss functions of MIPLMA and MIPLML are $\mathcal{L}=\mathcal{L}\_{\text{d}} + \lambda \mathcal{L}\_{\text{m}}$ and $\mathcal{L}=\mathcal{L}\_{\text{d}} + \lambda \mathcal{L}\_{\text{ml}}$, respectively. The results show MIPLMA's superiority, demonstrating that a direct adaptation of margin-based losses achieve worse performance.
>
> | Algorithm |   C-Row   |   C-SBN   | C-KMeans  |  C-SIFT   |
> | --------- | :-------: | :-------: | :-------: | :-------: |
> | MIPLMA    | .444±.010 | .526±.009 | .557±.010 | .553±.009 |
> | MIPLML    | .424±.006 | .505±.007 | .544±.009 | .545±.009 |
>
>
>
> > Applications need to be more convincing.
>
> In medical diagnostics, whole slide images (WSI) are high-resolution microscopy images of stained tissue slides. These images are extremely large, often reaching gigapixel size. Neural networks cannot learn an entire WSI due to their architecture and GPU limitations. Typically, WSIs are divided into smaller, lower-resolution tiles (multi-instances) to enable neural networks to learn meaningful features. In the future, we plan to collect higher-resolution WSI datasets to extend the MIPL applications.
>
>
>
> > The creation of the synthetic datasets is unclear.
>
> For the MNIST-MIPL, FMNIST-MIPL, and Birdsong-MIPL datasets, we can control the distribution of true instances. The SIVAL-MIPL dataset, derived from the classical MIL dataset SIVAL, is created by adding false positive labels to each multi-instance bag. We calculate and report the proportions of positive instances in each bag, including their maximum, minimum, mean, and median values. This detailed distribution information will be included in the revised manuscript.
>
> |         | MNIST-MIPL | FMNIST-MIPL | Birdsong-MIPL | SIVAL-MIPL |
> | ------- | :--------: | :---------: | :-----------: | :--------: |
> | Maximum |    9.1%    |    9.1%     |     10.0%     |   90.62%   |
> | Minimum |    7.0%    |    7.0%     |     6.9%      |   3.12%    |
> | Mean    |    8.0%    |    8.0%     |     8.3%      |   25.6%    |
> | Median  |   8.11%    |    8.11%    |     8.3%      |   21.9%    |
>
>
>
> > The comparison should be made using the same feature extractors.
>
> For the MNIST-MIPL and FMNIST-MIPL datasets, we use a two-layer CNN and a fully connected layer to extract instance-level features, which are then aggregated into a unified representation using the attention mechanism. The feature extractors for MIPLMA, ELIMIPL, and DEMIPL are consistent. However, the PLL methods cannot directly handle multi-instance bags or individual instances in MIPL because they lack of an aggregation mechanism to yield bag-level features and the candidate label sets are unknown during training. Therefore, it is not feasible to apply the same feature extractors across MIPL and PLLmethods.
>
>
>
> References:
>
> [B] Xu et al. Progressive purification for instance-dependent partial label learning. ICML 2023.

---

> ### Author Response · Authors · 2024-08-12
> **Looking Forward to Your Feedback**
>
> Dear Reviewer H7jM,
>
> Thank you for your thoughtful and detailed feedback, which has greatly improved our manuscript. We have carefully addressed your concerns by providing additional results on false positives in the SIVAL-MIPL dataset, comparing our method with the POP PLL method, and designing a variant of MIPLMA. We also included explanations about MIPL applications and the creation of synthetic datasets. Should any issues remain unresolved, we are happy to discuss them further.
>
> Best regards,
>
> The Authors

---

> > ### Comment · Reviewer_H7jM · 2024-08-12
> > **Post rebuttal response**
> >
> > The author response addresses some of my concerns and I have updated my rating accordingly. However, the last point is still unclear. Line 200-201, are misleading, why limit to linear or MLP backbone. I see no difficulty in a using a resnet backbone for instance (for example LWS based algorithm do use ResNet backbone in the experiments). And even if the authors are using pre-trained backbones, the adaption and fine-tuning is certainly possible.

---

> ### Author Response · Authors · 2024-08-12
> **Further Explanation on the Feature Extractor**
>
> Thank you for your feedback and for updating your rating.
>
> We acknowledge that various PLL methods, such as POP [B], PRODEN [C], and LWS [D], use different backbones like linear models, MLP, and ResNet depending on the dataset. For example, PRODEN and LWS mainly use linear and MLP models for MNIST and FMNIST datasets, while ResNet and convNet are employed for CIFAR-10 dataset. In our experiments, LeNet and ResNet-34 performed worse than our two-layer CNN on the MNIST-MIPL and FMNIST-MIPL datasets. This is likely due to the relatively simple nature of the features in these datasets, where deeper networks like ResNet may not provide significant advantages in weakly supervised scenarios. Additionally, the Birdsong-MIPL and SIVAL-MIPL datasets are preprocessed tabular data, which makes ResNet unsuitable for these datasets.
>
> For the C-R34-16 and C-R34-25 datasets, we used ResNet-34 as the feature extractor for both MIPLMA and the compared PLL algorithms. The table below presents the results of these algorithms, where Mean and MaxMin represent the aggregation strategies used to transform instance-level features into bag-level features. The results indicate that MIPLMA outperforms the compared PLL algorithms, even with the same feature extractor. We will include more detailed experimental results in the revised manuscript. Additionally, we plan to collect higher-resolution WSI datasets and the use of pre-trained backbones, including large language models, for better feature learning.
>
> | Algorithm       |   C-R34-16    |   C-R34-25    |
> | --------------- | :-----------: | :-----------: |
> | MIPLMA          | **.631±.008** | **.685±.011** |
> | LWS (Mean)      |   .593±.018   |   .614±.021   |
> | LWS (MaxMin)    |   .462±.014   |   .460±.016   |
> | PRODEN (Mean)   |   .537±.015   |   .585±.019   |
> | PRODEN (MaxMin) |   .434±.010   |   .444±.012   |
> | POP (Mean)      |   .549±.014   |   .591±.014   |
> | POP (MaxMin)    |   .450±.009   |   .462±.015   |
>
> References:
>
> [B] Xu et al. Progressive purification for instance-dependent partial label learning. ICML 2023.
>
> [C] Lv et al. Progressive identification of true labels for partial-label learning. ICML 2020.
>
> [D] Wen et al. Leveraged weighted loss for partial label learning. ICML 2021.

---

### Author Rebuttal · Authors · 2024-08-07

We sincerely thank the reviewers for their time and valuable feedback on our submission.

As we cannot resubmit the paper during the rebuttal period, we have attached a one-page PDF with additional experimental results for the new PLL algorithm POP on both benchmark and real-world datasets, results with varying temperature decay rates, and visualizations of attention scores and bag-level features. We will include these details in the revised manuscript.

Please let us know if further details or explanations are needed.

---

### Decision · Program_Chairs · 2024-09-25

**Decision:**

Accept (poster)

**Comment:**

The paper introduces MIPLMA, a Multi-Instance Partial Label algorithm that aims to maximize margins in both instance and label spaces. In instance space, it leverages a temperature parameter in the attention mechanism to increase the gap between attention scores for positive and negative instances. In label space, it employs a margin loss that combines two objectives: maximizing the margin between the highest predicted probability in the candidate label set versus the non-candidate label set, and minimizing the variance of these margins across bags. Empirical results on four benchmarks and one real world dataset show satisfactory performance compared with existing methods.

The reviewers appreciated the margin based formulation of the problem and found the empirical results to be promising. The technical novelty is on the low side but overall the paper presents a well executed study showing the benefits of different margin objectives for MIPL problems.